

# Cross-polar transport and scavenging of Siberian aerosols containing black carbon during the 2012 ACCESS summer campaign

Jean-Christophe Raut[1], Louis Marelle[1,a], Jerome D. Fast[2], Jennie L. Thomas[1], Bernadett Weinzierl[3,4,5], Katharine S. Law[1], Larry K. Berg[2], Anke Roiger[3], Richard C. Easter[2], Katharina Heimerl[3,4], Tatsuo Onishi[1], Julien Delanoe[1], and Hans Schlager[3]

[1]LATMOS/IPSL, UPMC Univ. Paris 06 Sorbonne Universités, UVSQ, CNRS, Paris, France
[2]Pacific Northwest National Laboratory, Richland, WA, USA
[3]Deutsches Zentrum für Luft- und Raumfahrt (DLR), Institut für Physik der Atmosphäre, Oberpfaffenhofen, Germany
[4]Ludwig-Maximilians-Universität, Meteorologisches Institut, Munich, Germany
[5]University of Vienna, Aerosol Physics and Environmental Physics, Vienna, Austria
[a]Now at Center for International Climate and Environmental Research, Oslo, Norway

*Correspondence to:* jean-christophe.raut@latmos.ipsl.fr

**Abstract.** During the ACCESS airborne campaign in July 2012, extensive boreal forest fires resulted in significant aerosol transport to the Arctic. A 10 day episode combining intense biomass burning over Siberia and low-pressure systems over the Arctic Ocean resulted in efficient transport of plumes containing black carbon (BC) towards the Arctic, mostly in the upper troposphere ($6 - 8$ km). A combination of in situ observations (DLR Falcon aircraft), satellite analysis and WRF-Chem

simulations are used to understand the vertical and horizontal transport mechanisms of BC with a focus on the role of wet removal. Between the northwestern Norwegian coast and the Svalbard archipelago, the Falcon aircraft sampled plumes with enhanced CO concentrations up to 200 ppbv and BC mixing ratios up to $25 \, \text{ng kg}^{-1}$. During transport to the Arctic region, a large fraction of BC particles are scavenged by two wet deposition processes, namely wet removal by large-scale precipitation and removal in wet convective updrafts, with both processes contributing almost equally to the total accumulated deposition of

BC. Our results underline that applying a finer horizontal resolution (40 instead of 100 km) improves the model performance, as it significantly reduces the overestimation of BC levels observed at a coarser resolution in the mid-troposphere. According to the simulations at 40 km, the transport efficiency of BC ($\text{TE}_{\text{BC}}$) in biomass burning plumes is about 60%, which is impacted by small accumulated precipitation along trajectory (APT) (1 mm). In contrast $\text{TE}_{\text{BC}}$ is very small ($< 30\%$) and APT is larger ($5 - 10$ mm) in plumes influenced by urban anthropogenic sources and flaring activities in Northern Russia, resulting in

transport to lower altitudes. $\text{TE}_{\text{BC}}$ due to grid scale precipitation is responsible for a sharp meridional gradient in the distribution of BC concentrations. Wet removal in subgrid parameterized clouds (cumuli) is the cause of modeled vertical gradient of $\text{TE}_{\text{BC}}$, especially in the mid-latitudes, reflecting the distribution of convective precipitation, but is dominated in the Arctic region by the grid-scale wet removal associated with the formation of stratocumulus clouds in the PBL that produced frequent drizzle.



## 1 Introduction

The Arctic region is particularly sensitive to environmental change, as it is predicted to warm faster than any other region (Manabe et al., 1992). The summertime extent of sea-ice has decreased significantly in recent decades (e.g., Lindsay et al., 2009). A warmer Arctic may lead to a substantial expansion of resource extraction and seasonal shipping traffic due to improved sea

access (Corbett et al., 2010; IPCC, 2013). In addition to long-lived greenhouse gas-induced warming and feedbacks, Arctic warming may also be caused by shorter-lived climate forcing agents (Tomasi et al., 2007; Law and Stohl, 2007; Quinn et al., 2008; AMAP, 2015), since aerosols and ozone perturb the radiative balance of the Arctic directly by absorbing radiation (Shindell and Faluvegi, 2009), and indirectly due to aerosol effects on clouds properties leading to increases in shortwave scattering efficiency and IR emissivity of Arctic clouds (Garrett and Zhao, 2006; Lubin and Vogelmann, 2006; Zhao and Garrett, 2015).

Black carbon (BC) is only a minor contributor to aerosol mass but is clearly a significant short-lived climate forcing agent in the atmosphere and when it is deposited onto snow and ice surfaces, reducing their albedo due to multiple scattering in the snowpack and the much larger absorption coefficient of BC than ice (Warren and Wiscombe, 1980; Hansen and Nazarenko, 2004; Jacobson, 2004), modifying snow grain size and driving changes in snow melt and surface temperature (Flanner et al., 2007, 2009; Jacobson, 2010).

Arctic aerosol research has predominantly focused on winter and spring while fewer measurement campaigns have been performed during summer when concentrations are generally smaller (Law and Stohl, 2007). More attention has been given to surface measurements, whereas only very few summertime airborne studies (e.g. POLARCAT, ARCTAS) of aerosol chemical composition have been carried out (Schmale et al., 2011). During summer, inefficient long-range transport reveals that summer-

time aerosol originates from the marginal Arctic (sea-ice boundary and boreal forest regions (Warneke et al., 2010)) and from episodic transport events, caused by rapid transport from heavily polluted but varied source types and regions, and producing dense haze layers aloft that rarely reach surface monitoring sites (Browse et al., 2012; Pierro et al., 2013; Ancellet et al., 2014; Law et al., 2014). Some studies (Brock et al., 1989; Stohl, 2006; Paris et al., 2009) identified boreal and temperate forest fires, especially Siberian fires, as a significant source of BC during the summer. Boreal forest fires may be the dominant source of

BC for the Arctic during intense fire years. Smoke from biomass burning tends to be highly stratified but mostly present at higher altitudes ($3 - 10$ km) because forest fire emissions can be injected well above the boundary layer during the flaming phase and even into the lower stratosphere by pyrocumulus clouds (Lavoué et al., 2000). Air masses can also be uplifted during transport to the Arctic in Warm Conveyor Belt (WCB) of synoptic low pressure systems, which have been found to be a dominant mechanism in vertically redistributing pollution (Browning and Monk, 1982; Cooper et al., 2002). Climatological

trajectory analyses have shown that northeastern China and Russia have a large frequency of WCB events (Eckhardt et al., 2004; Madonna et al., 2014). During the International Polar Year (IPY) in 2008, different studies using backward trajectory and lagrangian particle dispersion model analyses have highlighted WCB as an important pathway to transport pollution into the polar tropopause region during summer (Roiger et al., 2011; Sodemann et al., 2011; Matsui et al., 2011). These studies identified several Asian pollution plumes transported to the Arctic after a strong uplift in WCBs located over the Russian east-





coast and subsequently transported to the Arctic. Franklin et al. (2014) and Taylor et al. (2014) have documented the impact of wet removal in Canadian biomass burning plumes and confirmed that wet deposition is the dominant mechanism for BC removal from the atmosphere and consequently determining its lifetime, its atmospheric burden, and affecting vertical profiles of number and mass concentration.

State-of-the-art global chemical transport models have large discrepancies and spread in the prediction of Arctic BC, including incorrect seasonality and order-of-magnitude errors (Shindell et al., 2008; Koch et al., 2009; Lee et al., 2013; Schwarz et al., 2013; AMAP, 2015). Schwarz et al. (2010) showed that uncertainties in modeling of aerosol physical and chemical processing are extremely important, with lesser roles for emissions and dry transport. One of the main causes of these differences is large uncertainties in the representation of wet removal of BC adopted in aerosol models (Vignati et al., 2010; Bourgeois and Bey, 2011), but a quantitative understanding of this process is still limited (Browse et al., 2012). More recently, Breider et al. (2014) compared GEOS-Chem model results with mean vertical profiles of BC from the ARCTAS summer campaign. The observations in July were less than $25\,\mathrm{ng\,m^{-3}}$ throughout the vertical profile, but the model strongly overestimated the concentrations at high altitudes due to long-range transport of deep convective outflow from Asian pollution (summer monsoon) and Siberian fire plumes. This overestimation in remote regions confirms previous results from Koch and Hansen (2005). Wang et al. (2014) used a more recent version of this model, increasing the scavenging efficiency of BC in deep convection, but they also noticed a strong overestimation of BC in the mid-troposphere. The over prediction of BC concentrations during the summer might be due to the inability models to produce rain by convective clouds/fog that readily form during the summer in the Arctic (Sharma et al., 2013). These results reinforce the fact that aerosol and cloud physical and chemical processing (removal, oxidation and microphysics) is the primary source of uncertainty in modelling aerosol distributions in the Arctic (Eckhardt et al., 2015).

During the summer of 2012, extensive boreal forest fires occurred both in western and in eastern Siberia (in the Yakutsk region), influencing the atmospheric composition in the Arctic. These fires were associated with a hottest summer on record over Siberia, leading to very high emission rates of a number of trace species and aerosols (Ponomarev, 2013). In this paper, we focus on a case of pollutant transport from Asia and Russia across the Arctic because they may ultimately lead to in the transport of pollution to Europe and North America. During the ACCESS campaign, the aircraft sampled plumes representative of cross-polar transport reaching the Norwegian coasts. Combining an analysis of aircraft measurements, satellite observations, lagrangian trajectories and simulations from a regional online chemistry transport model (CTM), we investigate the horizontal spatial and vertical structure of BC-containing air masses, the factors affecting this transport, including the meteorological situation and the dry and wet deposition processes and we assess the relative contribution of anthropogenic and biomass burning sources to the BC concentrations. This synergy of methods gives good accuracy on short time scales for source regions with close proximity to the Arctic and enables us to treat pathway-dependent aerosol removal and quantify contributions from distant sources. Section 2 describes the ACCESS campaign, as well as the modeling tools used in this study. Model output is validated against ACCESS observations in terms of meteorological and chemistry variables, aerosol optical depth (AOD), precipitation and cloud structures (Sect. 3). The export processes, the identification of plumes and the transport pathways are





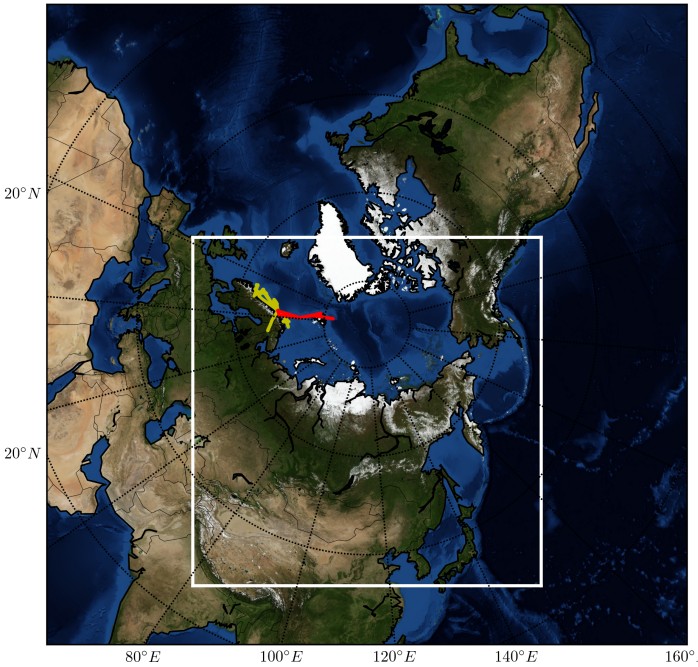

**Figure 1.** Map of the WRF-Chem domain (white) and the flights conducted as part of ACCESS used to evaluate the model. The Falcon-20 flights are in yellow. Flights of the 17 July are highlighted in red.

described in Sect. 4. Finally, Sect. 5 investigates the contribution of the various deposition processes and the transport efficiency of BC reaching the Norwegian coasts.

## 2 Methodology

### 2.1 ACCESS campaign

Roiger et al. (2015) have recently given a detailed overview of the ACCESS aircraft campaign. Therefore we only describe it briefly here. This campaign was based out of Andenes, Norway (69.3°N, 16.1°W), in July 2012 and included 14 flights over the northwestern coast of Norway using the DLR Falcon-20 aircraft (Fig. 1). Most of the flights were devoted to studying the impacts of local pollution sources (ships, oil and gas extraction) on Arctic atmospheric composition (Marelle et al., 2016), but pollution from remote sources was also measured during some specific flights in the middle and upper troposphere (Roiger et al., 2015). In this paper, we focus on all flights but describe in particular the two flights that both occurred on 17 July (namely flights 17a and 17b), which were specifically dedicated to probe father into the Arctic in order to study plumes transported from boreal and Asian sources (Fig. 1). IASI (Infrared Atmospheric Sounding Interferometer)-retrieved CO total columns as well as



global trace gas forecasts from the Monitoring Atmospheric Composition and Climate (MACC) were used during the campaign to target such flights in order to identify polluted air masses in the mid and upper troposphere (Roiger et al., 2015). In addition to the suite of meteorological instruments, the Falcon-20 was equipped with trace gas ($NO_x$, $SO_2$, $O_3$, and CO) and aerosol instrumentation including BC.

BC mass mixing ratios were measured using a single-particle soot photometer (SP2, Droplet Measurement Technologies, Boulder, CO, USA) based on laser-induced incandescence. The instrument measures the refractory black carbon (rBC; hereafter referred to as BC) on a single-particle basis (Schwarz et al., 2006). Particles are drawn through a high intensity 1064 nm Nd:YAG laser which heats BC-containing particles to incandescence. The individual particle masses of the BC cores of BC-

containing particles were averaged over 10 s intervals to obtain mass mixing ratios of BC. Since the SP2 instrument detected BC cores only in the size range of $80 - 470$ nm, assuming a density of $1.8\,\mathrm{g\,cm^{-3}}$ (Moteki et al., 2010), the derived mass mixing ratios were scaled by a factor to account for particles outside the detection range (as in e.g. Schwarz et al. (2006)) to obtain the total mass mixing ratio. The factor was derived from the average mass size distribution of BC cores separately for each flight and ranged from $1.1$ to $1.55$, depending on the encountered aerosol type. Absolute uncertainty of BC particle mass

is within $10\%$, the uncertainty of the derived total BC mass mixing ratio is about $30\%$.

## 2.2 Modeling strategy

### 2.2.1 WRF-Chem model set-up

In order to study the transport and processing of aerosols, regional chemical transport simulations are performed using the

Weather Research and Forecasting model with chemistry (WRF-Chem Version 3.5.1) (Grell et al., 2005). WRF model is a fully compressible meso-scale meteorological model designed for research experiments as well as operational weather forecasts (Skamarock and Klemp, 2008). The chemistry component (Grell et al., 2005) contains a number of modules for simulating aerosol direct and indirect effects and chemistry processes, which is fully consistent with the meteorological component (Fast et al., 2006). The different parameterizations and options used for the WRF-Chem simulations are summarized in Table 1.

Land surface processes are resolved using the Noah Land Surface model scheme with 4 soil layers (Ek et al., 2003). Planetary Boundary Layer (PBL) processes are parameterized according to the Mellor-Yamada-Janjic local scheme (Janjic, 1994). It is coupled to a surface layer based on Monin-Obukhov with Zilitinkevich thermal roughness length and standard similarity functions from look-up tables. We use the Morrison 2-moment (Morrison et al., 2009) microphysics scheme, which contains hydrometeor classes for water vapor, cloud water, rain, cloud ice, snow, and graupel. To represent the effects of subgrid-scale

convective processes on the grid variables, we use the Kain-Fritsch convective implicit parameterization (Kain, 2004), which has been recently improved by Berg et al. (2013) (details below). The shortwave and longwave radiation schemes are from the RRTMG model (Iacono et al., 2008). The Fast-J photolysis scheme (Wild et al., 2000) is coupled with hydrometeors as well as



**Table 1.** Parameterizations and options used for the WRF-Chem simulations

| Physical process | WRF-Chem option |
|---|---|
| Planetary boundary layer | Mellor-Yamada-Janjic (Janjic, 1994) |
| Surface layer | Monin-Obukhov |
| Land surface | Noah Land Surface model (Ek et al., 2003) |
| Microphysics | Morrison 2-moment (Morrison et al., 2009) |
| Shortwave radiation | RRTMG (Iacono et al., 2008) |
| Longwave radiation | RRTMG (Iacono et al., 2008) |
| Cumulus parameterization | Kain-Fritsch-CuP (KFCuP) (Berg et al., 2013, 2015) |
| Photolysis | Fast-J (Wild et al., 2000) |
| Gas phase chemistry | CBM-Z (Zaveri and Peters, 1999) |
| Aerosol model | MOSAIC 8 bins with aqueous chemistry (Zaveri et al., 2008) |

aerosols.

The model domain, shown in Figure 1, has a 40 km horizontal grid resolution on a polar stereographic grid and a distance of 8800 km in both horizontal directions. There are 50 vertical levels, and the lowest vertical grid spacing is about 25 m.

The model top is set to 50 hPa using a model time step of 180 s. The simulation begins on 00 UTC 4 July 2012 (after 3 days spinup) and ends on 00 UTC 21 July 2012. Initial and lateral boundaries of the meteorological fields are provided by NCEP (National Centers for Environmental Prediction) Global Forecast System (GFS) Final Analysis (FNL) 1° resolution data, which has 26 pressure levels and is updated every 6 hours. The simulations are also initialized with NCEP-archived 0.5° sea-surface temperatures fields (updated every 6 hours) and sea ice data (updated every day). The modeled horizontal wind

components, atmospheric temperature and humidity are nudged towards the 6-hourly NCEP/FNL reanalysis data (Stauffer and Seaman, 1990) above the PBL. The chemistry and aerosol species are initialized with reanalysis data from the offline global chemical transport model MOZART-4 (the Model for Ozone and Related Chemical Tracers) (Emmons et al., 2010). Spatially and temporally (6-hourly) varying chemical boundary conditions are also provided by MOZART-4. MOZART-4 data has a 2.5° horizontal resolution and uses regional and global anthropogenic and natural emissions inventories for chemistry and aerosol

species. The MOZART-4 reanalysis fields, provided by NCAR/NESL, contain 28 vertical levels with a maximum height at the 2 hPa pressure level.

In this study, the MOSAIC (Model for Simulating Aerosol Interactions and Chemistry) aerosol model (Zaveri et al., 2008) coupled with the CBM-Z (Carbon Bond Mechanism) photochemical mechanism (Zaveri and Peters, 1999) is used. MOSAIC

simulates a wide variety of aerosol species (sulfates, nitrates, ammonium, sea salt, dust, BC, organics, and others) using a sectional approach to represent aerosol size distributions by dividing up the size distribution for each species into several





size bins (8 in this study) and assumes that the aerosols are internally mixed in each bin. The MOSAIC aerosol scheme includes physical and chemical processes of nucleation, condensation, coagulation, aqueous phase chemistry, and water uptake by aerosols. It is also coupled to the microphysics and cumulus schemes, and to the shortwave and longwave radiation schemes.

Aerosol dry deposition is modeled using the "Resistance in Series Parameterization" (Wesely and Hicks, 2000), following the approach of Binkowski and Shankar (1995). Wet removal of aerosols by grid-resolved stratiform clouds/precipitation includes in-cloud removal (rainout) and below-cloud removal (washout) by rain, snow, graupel by Brownian diffusion, interception, and impaction mechanisms, following Easter et al. (2004) and Chapman et al. (2009). Aerosol-cloud interactions were included in the model by Gustafson et al. (2007) for calculating the activation and re-suspension between dry aerosols and cloud droplets.

Furthermore, Berg et al. (2015) have modified WRF-Chem to include treatments of a number of factors and processes important for accurately representing aerosol and trace gases within sub-grid-scale convective clouds, including fractional coverage of active and passive clouds, vertical transport, activation and resuspension, wet removal, and aqueous chemistry for cloud-borne particles. This new treatment, coupled to MOSAIC aerosol scheme, is important for including additional realism in regional-scale modeling studies that require the use of cumulus parameterizations when investigating the cloud effects on aerosol within

parameterized shallow and deep convection, and aerosol effects on cloud droplet number.

### 2.2.2 Emissions

The fire emissions inventory has been estimated using the Fire INventory from NCAR (FINN-v1) (Wiedinmyer et al., 2011). FINN provides emissions on a per fire basis based on fire count information from the MODIS (Moderate Resolution Imaging

Spectrometer) instrument. One of the most important aspects of simulating wildfire plume transport is the height at which emissions are injected. WRF-Chem contains an integrated one-dimensional plume rise model to determine the appropriate injection layer (Freitas et al., 2007; Grell et al., 2011; Sessions et al., 2011). The plume rise model accounts for thermal buoyancy associated with fires and local atmospheric stability, and is applied to distribute the fire emissions vertically. Anthropogenic emissions were taken from the Hemispheric Transport of Air Pollution version 2 (HTAPv2) $0.1° \times 0.1°$ inventory

(http://edgar.jrc.ec.europa.eu/htapv2/index.php?SECURE=123). Volatile organic compounds (VOCs) are given as a bulk VOC mass and distributed into CBM-Z emission categories assuming the speciation reported by Murrells et al. (2010). Time profiles are applied to anthropogenic emissions to account for the daily and weekly cycle of each emission sector (van der Gon et al., 2011). Soil-derived (dust) and sea salt aerosol emissions are incorporated into WRF-Chem (Shaw et al., 2008), and biogenic emissions are calculated online following the Model of Emissions of Gases and Aerosols from Nature (MEGAN) (Guenther,

30   2006).



**Table 2.** List of WRF-Chem simulations performed

| Name | Description |
| --- | --- |
| BASE | Baseline simulation at 40 km |
| NoFire | Run at 40 km without biomass burning emissions |
| NoAnthro | Run at 40 km without anthropogenic emissions |
| NoFlr | Run at 40 km without flaring emissions |
| NoDry | Run at 40 km without dry deposition |
| NoWet | Run at 40 km without wet deposition in grid scale clouds |
| NoWetCu | Run at 40 km without wet deposition in parameterized cumulus clouds |
| NoWetAll | Run at 40 km without wet removal in any clouds |
| Run100 | Run at 100 km with the same vertical resolution as in the BASE run |

### 2.2.3 Model simulations

The fine-scale structure of the Arctic atmosphere (e.g. due to the thermal stratification) poses a major problem for atmospheric transport model simulations. Here we run the WRF-Chem on a polar-stereographic projection at relatively high horizontal (40 km) and vertical (50 levels up to 50 hPa) to adequately represent the actual structure of the Arctic atmosphere, avoiding an overly rapid diffusion and decay along the boundaries of advected plumes (Rastigejev et al., 2010). A total of nine simulations are conducted: a baseline WRF-Chem simulation (BASE) and eight sensitivity tests to investigate the effect of several processes and emission sources on BC transport to the Arctic: a run without biomass burning emissions (NoFire), a run without anthropogenic emissions (NoAnthro), a run without emissions from flaring activities (NoFlr), a run without dry deposition (NoDry), a run without wet deposition in grid scale clouds (NoWet), a run without wet deposition in parameterized cumulus clouds (NoWetCu), and a run without wet removal in any clouds (NoWetAll). An additional simulation (Run100) similar to the BASE simulation, but at a coarser resolution (100 km) is also performed to study the influence of horizontal resolution on the transport and deposition of carbonaceous aerosols. The list of the WRF-Chem runs performed is summarized in Table 2.

### 2.2.4 FLEXPART-WRF

We use the Lagrangian particle dispersion model FLEXPART-WRF (Fast and Easter, 2006; Brioude et al., 2013) to assess pollution transport and dispersion from individual sources, and to identify the origins of measured pollution plumes during the ACCESS campaign. FLEXPART-WRF is derived from the FLEXPART dispersion model (Stohl et al., 2005), but is driven by meteorological fields simulated by WRF. In this study, FLEXPART-WRF is run in backward mode to study source-receptor relationships: 10000 particles are released at a receptor point and are transported backward in time using the meteorological fields from the WRF-Chem simulation. Source-receptor relationships are quantified by calculating the potential emission



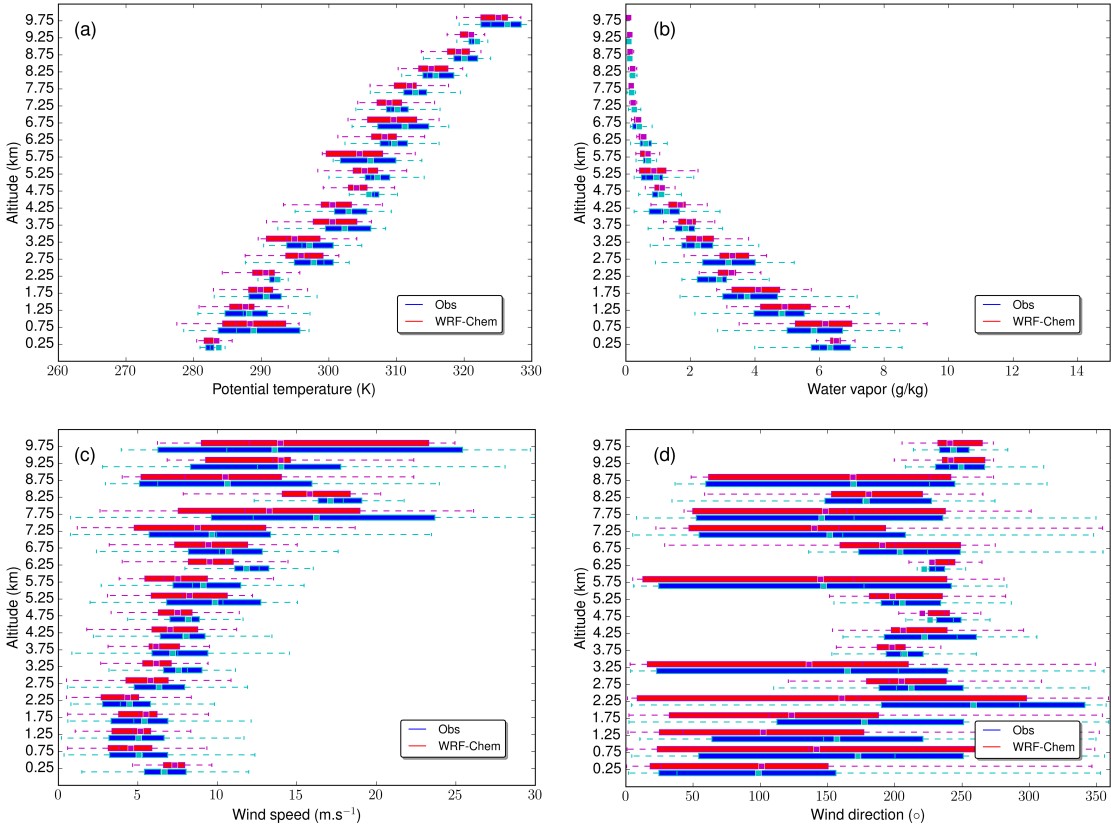

**Figure 2.** Vertical profiles of observed (blue) and modelled (red) (a) potential temperature (K), (b) water vapor mixing ratio $(\mathrm{g\,kg^{-1}})$, (c) wind speed $(\mathrm{m\,s^{-1}})$ and (d) wind direction (°) interpolated along the Falcon flight tracks. Boxes represent the interquartile range, vertical bars are the median values, squares show the means and whiskers show to the mininum and maximum of the data.

sensitivities (PES), which represent the amount of time spent by the particles in every grid cell. In this paper, meteorological variables are interpolated in time and space over trajectories from the WRF runs. FLEXPART-WRF is therefore useful to identify the origins and transport pathways of pollution plumes observed during the campaign and to derive precipitation and deposition efficiencies along those transport pathways (Sect. 5).

5 **3 Model validation**

**3.1 Meteorological parameters**

To evaluate the skill in the modelled weather and transport patterns predictions, we first compare ACCESS observations of temperature, relative humidity (RH), wind speed and wind direction to model results. The simulated meteorology is extracted every minute along the flight track using linear interpolation at the exact latitude, longitude, altitude and time using the model



**Table 3.** Performance statistics for meteorological variables and chemical species ($BC$, $CO$). For all flights, $R^2$, $MB$, $RMSE$ and $NMB$ represent the Pearson correlation coefficient, the mean bias, the root-mean-square error, and the normalized mean bias, respectively.

| Variable (unit) | $R^2$ | $MB$ | $RMSE$ | $NMB(\%)$ |
|---|---|---|---|---|
| Pressure (hPa) | 0.999 | 10.14 | 13.61 | 1.37 |
| Potential temperature (K) | 0.997 | -0.96 | 1.60 | -0.32 |
| Relative humidity (%) | 0.709 | 1.29 | 14.60 | 1.73 |
| Wind speed ($\mathrm{m\,s^{-1}}$) | 0.866 | -0.27 | 2.39 | -3.33 |
| Water vapor mixing ratio ($\mathrm{g\,kg^{-1}}$) | 0.924 | 0.15 | 0.61 | 3.67 |
| BC ($\mathrm{ng\,kg^{-1}}$) | 0.468 | 1.58 | 7.44 | 27.30 |
| CO (ppbv) | 0.524 | 1.47 | 17.63 | 1.49 |

output available at hourly intervals. The ensemble of observed and simulated data interpolated along the 14 flight tracks are then gathered to compute vertical profiles of key meteorological parameters influencing the long range chemical transport. The comparison with Falcon observations is shown in Figure 2. The predicted temperature is in very close agreement with airborne measurements, with only a small negative bias (about $-1$ K) at high altitudes. The modeled wind direction also matches the observations closely, with a slight positive normalized mean bias ($NMB < 6\%$) in the boundary layer. We note however that statistical performances for wind direction should be taken with caution as they are derived from a vector quantity. The simulated wind speed shows good agreement with a correlation coefficient of $R^2 \simeq 0.866$, although a slightly negative bias (lower than $3\%$) at altitudes above $6$ km. The water vapor mixing ratio and the resulting RH are also well reproduced by the model with high correlation coefficients (Table 3). Overall the WRF model shows appreciable skill in capturing the variability seen in the observations, suggesting that the large scale transport patterns and major aspects of flow conditions are well represented in the WRF simulations for this campaign.

### 3.2 Trace gases and aerosols

Figure 3 shows the vertical profiles of the CO and BC mixing ratios measured during the campaign and simulated by WRF-Chem (BASE and Run100 simulations). On average, the ACCESS measurements show enhanced BC and CO concentrations between 6 and 9 km of altitude. The two profiles are well correlated with maximum CO values of $200$ ppbv at $7-8$ km, associated with elevated BC values reaching $25$ $\mathrm{ng\,kg^{-1}}$. The corresponding mean values for CO and BC in this altitude range are $135$ ppbv and $12$ $\mathrm{ng\,kg^{-1}}$, respectively. The small underestimation in CO between 6 and 9 km is a common feature observed by most models (Emmons et al., 2015; AMAP, 2015) and is caused by chemical schemes leading to wrong OH and transport (Monks et al., 2015). A sensitivity test (not shown) based on HTAPv2 emissions versus ECLIPSE (Evaluating the CLimate and Air Quality ImPacts of Short-livEd Pollutants) inventory with much larger flaring emissions in northern Russia





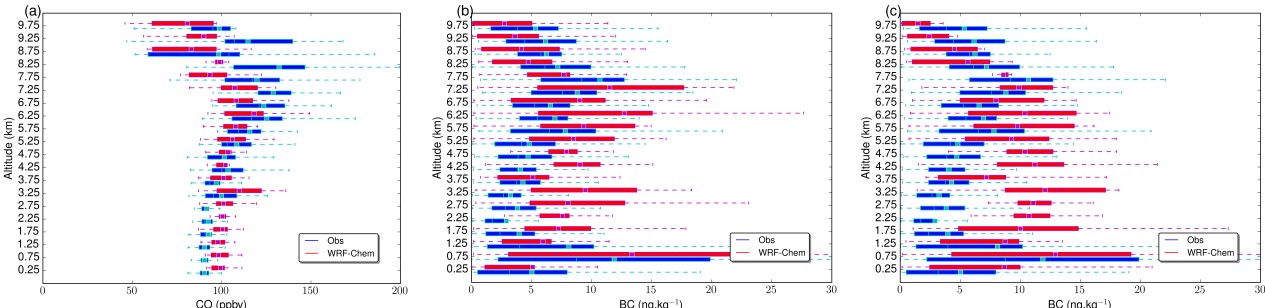

**Figure 3.** Vertical profiles of observed (blue) and modelled (red) mixing ratios of (a) CO (ppbv), (b) BC ($\mathrm{ng\,kg^{-1}}$) at a horizontal resolution 40 km (BASE), and (c) BC ($\mathrm{ng\,kg^{-1}}$) at a horizontal resolution 100 km (Run100) interpolated along the Falcon flight tracks. Boxes represent the interquartile range, vertical bars are the median values, squares show the means and whiskers show to the mininum and maximum of the data.

(Stohl et al., 2015) has suggested that the influence of flaring emissions in this area is insignificant. A noticeable increase in ozone concentrations is also clearly seen at 7 km, with a mean value of 135 ppbv (not shown). The vertical profiles of CO, BC and ozone demonstrate that, during summer 2012, transported pollution increased trace gases and aerosol concentrations in the middle and upper troposphere compared to typical clean background values (Roiger et al., 2015). The statistical performance

of the model in regards to the representation of key chemical species extracted along flight tracks are given in Table 3. The comparison of simulated CO ($NMB < 1.5\%$) against the Falcon flight observations showed that the model predictions are able to capture the magnitude and temporal variability seen in observed values. The model shows appreciable skill in capturing the vertical profile of BC, but overestimates the BC mixing ratio between 2 and 3 km of altitude. The resulting normalized mean bias on BC (27.3%) is lower than the error on the SP2 instrument ( 30%) and much lower than biases reported for most models

on the Arctic region (Eckhardt et al., 2015; AMAP, 2015). The mean profile of BC sampled during the ACCESS campaign is of the same order of magnitude as those reported by Breider et al. (2014) (25 $\mathrm{ng\,m^{-3}}$ in July 2008) and by Schwarz et al. (2013) during the HIPPO campaign at similar latitudes (between $60°$ and $80°$N), but in the remote Pacific region ($10-20$ $\mathrm{ng\,kg^{-1}}$). The noticeable increase of CO, an incomplete combustion product, and BC between 6 and 9 km of altitude in this study, associated with higher ozone mixing rations, points towards a regular transport of forest fire/biomass burning emissions

to northern Norway, providing us with the opportunity to study to chemical and aerosol composition in these plumes. This is discussed further in Sect. 4 (source attribution).

### 3.3 Model resolution

The result of the Run100 simulation is also shown in Fig. 3. The corresponding BC vertical profile shows the same enhance-

ment between 6 and 9 km as the BASE run, but it also clearly highlights a strong overestimation ($\times3$) in the mid-troposphere. This suggests that, at a coarser resolution, the model is unable to resolve the fine structure of plumes transported in altitude,




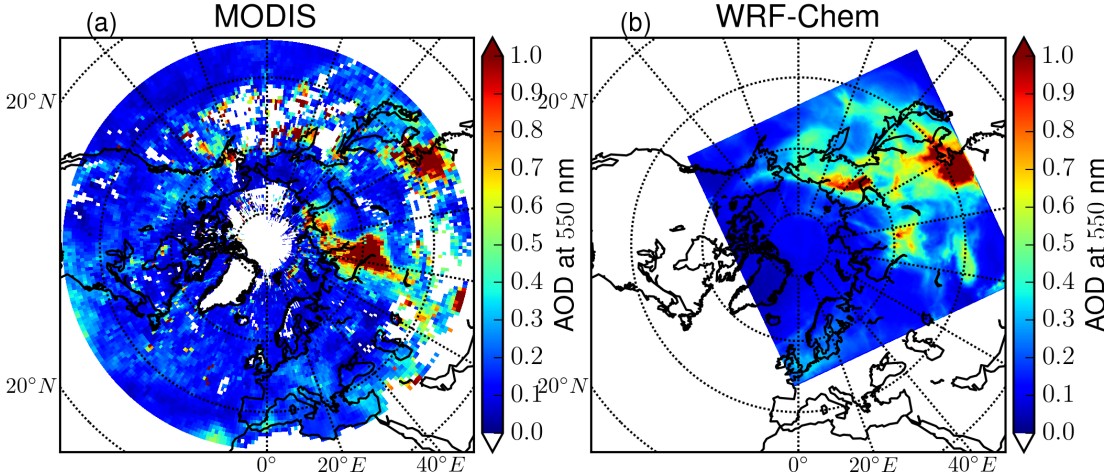

**Figure 4.** Aerosol optical depth at 550 nm (a) measured by MODIS instrument aboard Aqua and (b) simulated by a Mie code in the WRF-Chem model, averaged between 4 and 21 July 2012. White areas in (a) indicate the presence of clouds, preventing the AOD retrieval.

illustrates the difficulty to represent the cloud and precipitation structures and points for the need for improved representation of BC processing in global models and the necessity to compare models with measurements in Arctic regions. More generally, global models are commonly run at horizontal and vertical resolutions that are inadequate for representing the actual structure of the Arctic atmosphere, mostly because of computational limited resources. Schwarz et al. (2013) studied that the bias

between BC measurements and the AeroCOM model suite in remote regions. Global models always overestimated BC mass concentration, especially in the Arctic upper troposphere where the overestimation was about a factor of 13, suggesting that the aerosol lifetime and removal was not correctly in models. Model features that govern the vertical distribution and lifetimes of short-lived climate forcers in the Arctic atmosphere must still be improved (AMAP, 2015). Ma et al. (2013) showed that BC was better simulated with higher spatial resolution, and described that there is likely less wet removal at higher spatial resolution

since aerosols and clouds do not overlap as much. In our study, at a horizontal resolution of 40 km, the model-to-observations comparison indicates that the summertime transport pathways of pollution into the Arctic including the mid-latitude pollution transport and forest fire/biomass burning are well represented in the WRF-Chem model. At a finer resolution, the model is also able to better represent mean vertical motions associated with large-scale synoptic systems, i.e. conveyor belts, that lift PBL aerosols into the free troposphere. Resolution can therefore affect long-range transport, since it affects how high aerosols are

lifted into the free troposphere.

## 3.4  Aerosol optical depth

To evaluate the total aerosol column simulated by WRF-Chem during the ACCESS campaign, we calculate the aerosol optical depth (AOD) at 550 nm, defined as the integrated extinction coefficient over a vertical column of unit cross section, taking





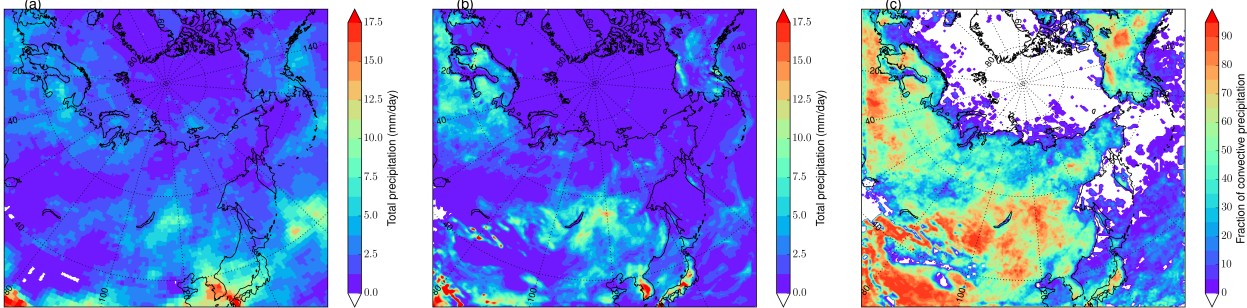

**Figure 5.** Average precipitation from period 4-21 July (a) obtained from $1°$ daily GPCP data and (b) simulated by WRF-Chem on the same grid. The fraction of convective precipitation is shown in panel (c).

into account the attenuation of the radiance by aerosol scattering and absorption. This is compared to the AOD retrieved by the MODIS instrument aboard Aqua satellite passing over the equator in the afternoon. The MODIS level 3 products used in this study are described by Hubanks et al. (2008). MODIS AOD fields at $550\,\text{nm}$ are retrieved from daily observations averaged over the period of the ACCESS campaign and taking into account missing observations primarily due to the presence of clouds

within the column. Fig. 4 shows average maps of observed and simulated AOD at $550\,\text{nm}$. Note that the AOD retrieved from MODIS above Terra satellite is very similar to that obtained from Aqua (not shown). In general, the model correctly represents the main features of the spatial AOD distribution, including the large values over northeastern China and western Siberia. In China, high AODs reaching 2.3 (1.8 in the model) are linked to high pollution levels in the PBL combined with strong dust episodes. Over western Siberia, the extended area with substantially enhanced values of AOD (up to 2.5) is due to large fires

(Ponomarev, 2013) and with a potential contribution from intense flaring activities where oil and gas resources are exploited (Stohl et al., 2013). The AOD is however underestimated by the model by $25\%$ over this latter region, due to the fact that the secondary organic formation is simplified in this current version of the model. The model also highlights a strong signal over the Yakutsk region (maximum AOD of 2.2), which is less visible on the MODIS images due to the continuous presence of clouds in July 2012. This is due to intense biomass burning plumes at this period (Ponomarev, 2013). The plumes extend

towards the East and above the Bering Strait. Other significant enhancements of modelled AOD ($0.4-0.5$) are also seen above northern India and above the Pacific Ocean, corresponding to the outflow area of China. The good performance to represent the main features of AOD distribution, especially in eastern Asia and western Siberia, gives a good confidence in the model to transport plumes to the Arctic. The underestimation above northern Russia indicates that the transport of plumes from that area should be considered with caution. Notwithstanding, due to the large amount of precipitation in this area (Sect. 3.5), the

potential impact of this source area is reduced.



### 3.5 Precipitation

The lofting of BC can occur isentropically at low RH or can be caused by rapid ascent and heavy precipitation (Matsui et al., 2011). Studying precipitation during transport is therefore crucial and it needs to be modeled correctly to understand the role of wet removal of aerosols. The reproducibility of precipitation in the WRF-Chem BASE simulation is assessed using daily GPCP (Global Precipitation Climatology Project) precipitation data. Here we use GPCP data (Version 1.2) available on a latitude-longitude grid with a resolution of $1°$, which is based on a combination of satellite observations and rain gauge measurements (Huffman et al., 2001). Model estimates of rainfall are interpolated every day to the GPCP $1°$ grid and then averaged between 4 and 21 July to give an assessment of the mean precipitation during the ACCESS campaign. The WRF-Chem simulation overall reproduces the spatial distribution of the observed precipitation reasonably well (Fig. 5), with highest precipitation intensity over South-East Asia (South Korea and Japan) with about $17 \, \mathrm{mm \, day^{-1}}$, high values over northern China, Mongolia and in the vicinity of Lake Baikal ($9-11 \, \mathrm{mm \, day^{-1}}$) or over the Pacific Ocean due to storm tracks and associated WCBs ($10-13 \, \mathrm{mm \, day^{-1}}$). Modest values are detected over Europe with $5-8 \, \mathrm{mm \, day^{-1}}$. The model nevertheless overestimates the precipitation intensity over southern Russia close to Lake Baikal and over Europe (by $20\%$) and underestimates values over the Pacific Ocean. Over Siberia and central Arctic the precipitation intensity is low ($1-3 \, \mathrm{mm \, day^{-1}}$). The highest values of rainfall are correlated with high convective precipitation fractions (Fig 5), reaching $60$ to $90\%$ over land. Over the oceans and northern Russia the main contribution is from non-convective precipitation (e.g. drizzle).

### 3.6 Cloud vertical distributions

Prediction of Arctic clouds (regional extent and heights) remains a challenge but is also crucial, since the vertical distribution of clouds as well as the state of cloud microphysics is important to understand how models assess the wet removal of aerosols. To evaluate the vertical distributions of clouds in our BASE simulation, we use the DARDAR-MASK v1 data set, which employs a combination of the CloudSat, CALIPSO (Cloud-Aerosol Lidar and Infrared Pathfinder Satellite Observations), and MODIS products (Delanoë and Hogan, 2010). To retrieve cloud phase properties, the algorithm uses the $94 \, \mathrm{GHz}$ radar reflectivity from CloudSat, the lidar backscatter coefficient at $532 \, \mathrm{nm}$ and vertical feature mask from CALIPSO, three MODIS infrared channels $8.55 \, \mathrm{mm}$, $11 \, \mathrm{mm}$, $12 \, \mathrm{mm}$ from MODIS, as well as ECMWF (European Centre for Medium-Range Weather Forecasts) thermo-dynamic variables. A range of categories is returned from the DARDAR-MASK: clear, ground, stratospheric features, aerosols, rain, supercooled liquid water, liquid warm, mixed-phase and ice. The algorithm also includes an uncertain classification, in regions where the radar and lidar signal are heavily attenuated or are missing. In this study, we use a simple DARDAR-MASK simulator for WRF-Chem enabling a clear comparison between the model and satellite observations. Mass mixing ratios of liquid water, ice crystals, rain, graupel and snow, as well as the temperature, are interpolated in time and space from the model along each CALIPSO or CloudSat track passing through the WRF-Chem domain during the simulation (4-21 July 2012). The observed and simulated masks are then temporally averaged at each altitude bin for each cloud phase category: uncertain, ice, mixed-phase (ice and supercooled water), liquid warm, supercooled water and rain. The result is a mean vertical profile for



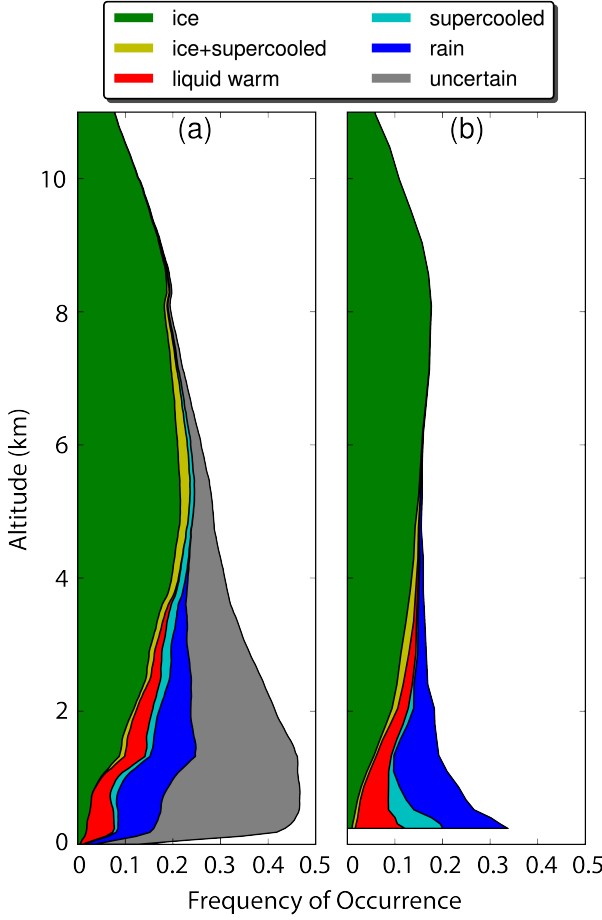

**Figure 6.** Mean vertical profiles of the different cloud phase categories (a) returned from the DARDAR-MASK and (b) simulated by the WRF-Chem model between 4 and 21 July.

each category and is represented in Fig. 6. Not surprisingly, the cloud fraction is more important in the lowest layers than in altitude, as the source of water in clouds is the evaporation from the Arctic Ocean and the humid soil. In the upper troposphere, the fraction of occurrence is 20% both in DARDAR and WRF-Chem, and clouds are composed only of ice crystals. This is directly liked to the temperature of clouds, below $-40°C$ at those altitudes. The ability of the model to represent those high

5  clouds is excellent. In the PBL, the cloud fraction is larger (between 20% and 35% if the uncertain category is not taken into account) and is dominated by liquid warm and rain. The main discrepancy between the model and the satellite observations can be ascribed to the uncertain cloud type. When the cloud type is classified as uncertain because the lidar signal is extinguished or too attenuated below optically thick clouds, the model generally predicts no cloud. The model slightly overestimates the fraction of clouds containing supercooled water in the PBL (by 2%) whereas this fraction is underestimated in the mid-

10  troposphere. The model also slightly underestimates the fraction of ice and supercooled droplets in the upper troposphere. If we



do not consider the uncertain class, the result of the WRF-Chem model shows appreciable skill in capturing the average vertical distribution of cloud phases. In particular, the vertical distribution of the liquid warm and rain categories is very well reproduced. This is very promising in simulating the wet deposition efficiency of aerosols in plumes transported to the Arctic region.

## 4 Source attribution of BC particles

The major objective of this section is to understand the vertical transport mechanisms of BC particles over Siberia and their transport pathways towards the Arctic in summer using the results from WRF-Chem calculations. First, we identify the major uplifting processes for BC particles and the major transport pathways for the export of BC from Siberia. In particular, we conduct analyses of the mass flux of BC and meteorological fields that play an important role in the transport and wet removal of BC. Second, we quantify the contributions of BC emitted from various Russian, Asian, and European regions to the outflow towards the Arctic.

### 4.1 Export processes

To identify the sources of pollution BC export to the Arctic and the mechanisms involved in the mean upward motion, we compute the horizontal poleward eddy heat flux at the 850 hPa level using the model (Fig. 7). The horizontal poleward eddy heat flux is calculated as $\overline{v'\theta'}$, where the overbar denotes time-averaging over the ACCESS period, $v'$ and $\theta'$ are the instantaneous deviations from the means of the meridian wind component and the potential temperature, respectively. This eddy heat flux is computed at each grid point of the WRF-Chem simulation. Four regions with large eddy heat transport can be identified during the ACCESS period. The values were large in northern central Europe, particularly Germany and Poland ($50 - 63°$N, $10 - 40°$E), over the West Siberian Plain ($60 - 70°$N, $40 - 80°$E) extending above the Barents Sea, over the Sakha (Yakutia) Republic ($65 - 70°$N, $110 - 150°$E) and finally over the Pacific storm track around northern Japan. Regions with large values of the horizontal poleward eddy heat flux generally illustrate the activity of migratory cyclones and may reflect strongly ascending moist airstreams (WCBs) in extratropical cyclones that rise from the PBL to the free troposphere (FT) on short time-scales (1 or 2 days) (Eckhardt et al., 2004). If these regions characterized by large poleward eddy flux are located in the vicinity of BC sources, they are likely to contribute to export pollution to the Norwegian Arctic, including areas sampled by the Falcon.

In Fig. 7, the spatial distribution of the mean upward BC mass fluxes averaged over the ACCESS period ($\overline{[BC]w}$), where $w$ denotes the vertical velocity) is also represented at the same altitude. Three regions with strong upward BC fluxes can be identified: central and northeastern China ($30 - 45°$N, $110 - 120°$E), south of the Krasnoyarsk region ($50 - 60°$N, $80 - 100°$E) and Yakutia ($60 - 68°$N, $125 - 150°$E). Those regions characterized by a strong ascent of BC mass fluxes are also co-located with areas presenting large values for the divergence of horizontal BC flux in the PBL, defined here as the $700 - 1000$ hPa layer (Fig. 7). The strong divergence regions of the horizontal BC fluxes within the PBL generally correspond to the large BC source regions and the divergences represent the horizontal transport of BC emitted from those source regions to the





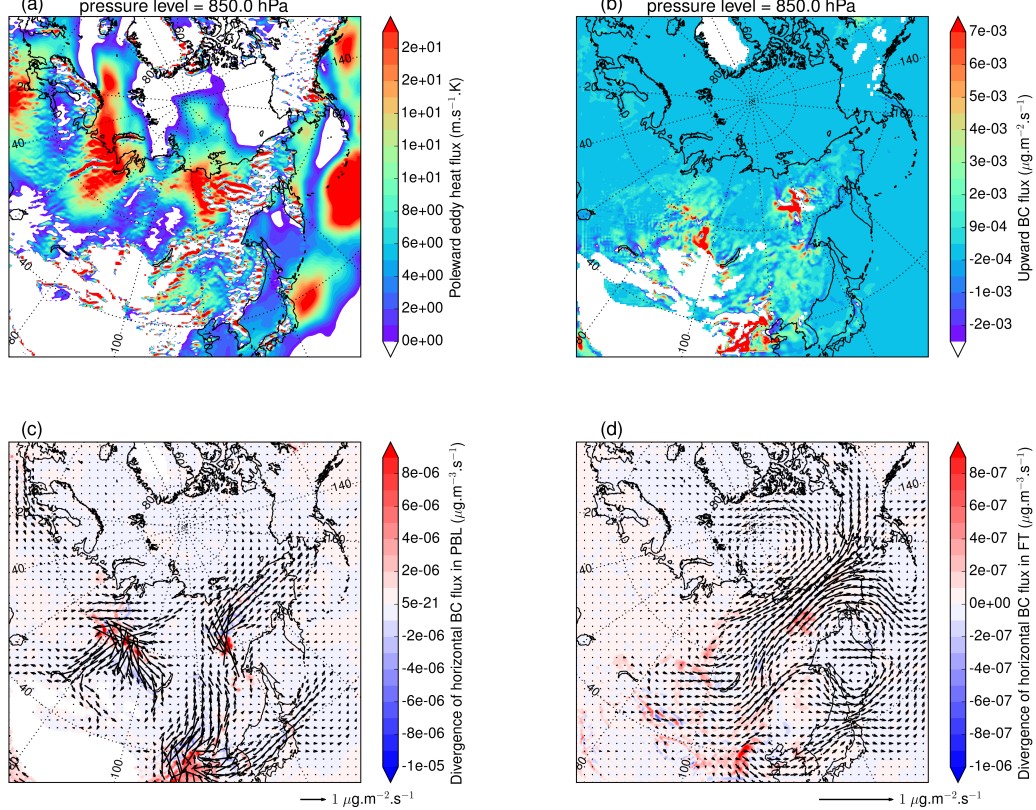

**Figure 7.** Time-averaged (a) horizontal poleward eddy heat flux $\overline{v'\theta'}$ at the 850 hPa level, (b) upward mass flux of BC $\overline{[BC]w}$ at the 850 hPa level, and divergence of the horizontal mass flux of BC integrated over (c) the PBL and (d) the FT during the ACCESS period in the WRF-Chem BASE simulation. Regions in (b) and (c) without data in white correspond to regions with high-altitude mountains (surface pressure below 850 hPa). Vectors representing the time-averaged horizontal mass flux of BC integrated over the PBL and FT during the ACCESS period are shown in panels (c) and (d) respectively, as well as the scaling near the lower right corner.

surrounding regions (Oshima et al., 2013). A good spatial correlation is also observed with the spatial distribution of AOD (Fig. 4) illustrating that the main emission areas present a significant fraction of carbonaceous aerosols. BC emitted from central and northeastern China is mostly of anthropogenic origin, whereas the two other regions (south of the Krasnoyarsk region and Yakutia) are co-located to intense biomass burning sources, explaining the strong values of the upward BC flux.

5   Aerosols and CO emitted from those source regions may potentially be exported towards the Arctic through the areas presenting large horizontal poleward eddy heat flux: the Sakha (Yakutia) Republic ($65-70°N$, $110-150°E$) and the Pacific storm track. It is confirmed with the time-averaged horizontal mass flux of BC integrated over the FT during the ACCESS period (Fig. 7). We also note that the source of BC identified in western Siberia is also partly located in the region where the Russian oil and gas flaring emissions are very high and along the main low-level pathway of air masses entering the Arctic (Stohl et al.,




2013). They may be exported towards the Arctic through the region with high value of $\overline{v'\theta'}$, extending above the Barents Sea. Anthropogenic emissions in Europe may also be exported to the Arctic but to a lower extent since the corresponding horizontal and vertical mass fluxes of BC are low in summer.

## 4.2 Plume identification

5  As indicated in Sect. 2.1, the two flights of 17 July (namely flight 17a in the morning and flight 17b in the afternoon) were specifically dedicated to plumes transported in altitude from boreal and Asian sources to the Arctic region. Data obtained during these flights is thus the most relevant for the study of BC transport to the Arctic. Figure 8 shows the vertical cross-sections of CO and BC extracted from the WRF-Chem model on the two flight tracks of 17 July. In situ measurements of these variables are also shown in Fig. 8. The predicted time-altitude cross-sections along the selected flight tracks help in distinguishing the horizontal and vertical variability of the CO and BC enhancements due to different source regions. The Falcon transected pollution plumes located between Andøya and Spitsbergen during the first flight and sampled the same plumes again during the second flight before returning to Andøya. The model simulates various CO enhancements during each flight period. Increases in BC mixing ratios are very well co-located with those CO enhancements. The PV (potential vorticity) vertical cross-section suggests that all those enhancements are in the troposphere: at the time of the flights, the stratospheric air masses are confined in regions with potential temperature larger than 310 K, with a fold between 09:30 and 13:30 UTC, bringing stratospheric intrusions down to 310 K. In the upper troposphere, the model predicts three periods of enhanced CO and BC during each flight, near the 310 K isentrope (altitude between 6 and 8 km). In this altitude range, when the Falcon crosses the plumes predicted by the model, the instruments also detect higher concentrations of CO and BC. WRF-Chem however underestimates the magnitude of the mixing ratios of carbon monoxide in the peaks, reaching 210 ppbv. This is mostly due to numerical dif-

20  fusion in the model, but also to the fact that the Falcon sampled the southern edge of a larger and more intense plume located farther north before returning to Andenes (Roiger et al., 2015). The underestimation of CO can also partly be ascribed to too low emissions or problems in modeling OH. This plume is shifted a bit towards the north in the model simulation, explaining the strong underestimation of the model at 09:15 UTC. In terms of BC, the agreement between WRF-Chem and the SP2 instrument is very good. Inside the plumes, values of $20 - 25 \, \mathrm{ng \, kg^{-1}}$ are reported, with peak values larger than $30 \, \mathrm{ng \, kg^{-1}}$. In the mid-troposphere, CO and BC concentrations are also significantly increased sporadically in the band delimited by the 290 and 300 K isentropes (between 2 and 4 km) with values reaching 180 ppbv and $30 \, \mathrm{ng \, kg^{-1}}$, respectively.

## 4.3 Contribution from the different sources

To understand the contributions of the different source emissions (anthropogenic, biomass burning and flaring) to the BC trans-

30  ported to the region of the flights, we use the difference between the BASE run and the NoAnthro, NoFire, NoFlr simulations. Model results from each run are interpolated in time and space to the position of the Falcon during the two flights on 17 July. The relative contribution of each source to total BC is obtained by dividing by the results derived from the BASE simulation. Figure 9 shows the time-altitude cross-sections of the relative contributions of the different emission sources. The two verti-



**Figure 8.** Vertical cross-sections of (a) and (b) CO mixing ratio, (c) and (d) BC mixing ratio, (e) and (f) PV along Falcon flight tracks on 17 July 2012 extracted from WRF-Chem between Andøya island and Svalbard archipelago. In situ measurements aboard the Falcon are shown in each panel with colored dots, using the same color scale. Black solid lines represent the dry isentropes between 290 K and 340 K.

cal cross-sections clearly underline that the mass mixing ratio of BC is strongly dominated by the fire contribution, which is







**Figure 9.** Time-altitude cross-sections of the relative contributions (%) of (a) and (b) anthropogenic, (c) and (d) biomass burning and (e) and (f) flaring emissions along flight tracks 17a (in panels (a), (c) and (e)) and 17b (in panels (b), (d) and (f)). The altitude of the two flights is indicated in magenta in each panel.

generally larger than 80% at all altitudes and times, and often larger than 90%. We note two exceptions to the dominance of



fires. In the vicinity of the Norwegian coast, i.e. just after take off from Andenes before 09:00 and on the way back to Norway after 14:00 UTC, a strong influence of anthropogenic sources (about $85\%$) is clearly identified between 2 and 4 km, and to a lesser extent in the PBL, where a weaker contribution of Norwegian flares is predicted ($40 - 70\%$). A second zone where the influence of fires is low (below $15\%$) is detected in the mid-troposphere ($2 - 5$ km) between 12:30 and 13:00 UTC. In this
region, the contribution of flaring emissions from Siberian oil exploration is dominant, about $80\%$.

In Sect. 4.2, we identified significant enhancements in CO and BC between 6 and 8 km. According to Fig. 9, these plumes originate from biomass burning sources. Heating of large Siberian fires can inject CO and BC into the free troposphere. The online plume-rise injection model in WRF-Chem is used to predict fire emission injection heights in our simulations (Grell
et al., 2011). The convective motions that occur with such wild fires increase the likelihood that emissions will be lofted to the faster winds of the free atmosphere. We should note that in this altitude range ($5 - 10$ km), the contribution of anthropogenic sources is weak (maximum $15\%$) but not zero. There is likely a small influence of the vertical transport by continental convective lifting over Asia in summer and frontal lifting associated with cyclones in the mid-latitudes. The transport pathways from the different source regions to the Arctic is discussed in Sect. 4.4. In the mid-troposphere ($2 - 4$ km), the CO and BC
concentrations have also increased due to the influence of fires, except in the vicinity of the northern Norwegian coast, where the impact of anthropogenic sources is larger. The flaring emissions play a very localized role in the area of the flights and are not due to any noticeable increase in CO or BC. This suggests that a large fraction of the aerosols emitted from flares has been removed by precipitation during transport, which is investigated in more detail in Sect. 5.

**4.4    Transport pathways**

We use FLEXPART-WRF in backward mode to study the origin and transport pathways of plumes and to provide insight into the WRF-Chem representation of BC. We identify four plumes originating from boreal fires on 17 July from Fig. 8 in the upper troposphere between 6 and 8 km and confirm them by in situ measurements from the aircraft. The center of those plumes has been found at 08:42, 09:39, 13:06, 14:10 UTC. Their dominant biomass burning origin has been suggested by Fig. 9. We
also consider two plumes predicted to be influenced by flaring emissions, in the PBL (0.7 km) at 08:05 UTC and in the mid-troposphere (3.7 km) at 12:35 UTC. Finally two anthropogenic plumes have been selected in the mid-troposphere ($3.2 - 3.5$ km), in the proximity of Andøya island at 08:30 and 14:45 UTC. For each selected plume, 10000 FLEXPART-WRF particles are released in a volume 40 km $\times 40$ km (horizontally) and 1 km (vertically). Since transport times are typically less than 10 days, each of the simulations is run backwards for 10 days to track the air mass origin over the source regions of interest.
Specifically, we use FLEXPART-WRF PES to study source-receptor relationships for air masses reported in Fig. 9.

Figure 10 shows the $0 - 20$ km column of FLEXPART-WRF PES integrated for 10 days for the first fire plume (released at 08:42 UTC on 17 July), together with the altitude and BC concentration retrieved from the model and interpolated along each of the 10000 trajectories. A vertical cross-section of BC interpolated along the plume centroid location is also shown. We note



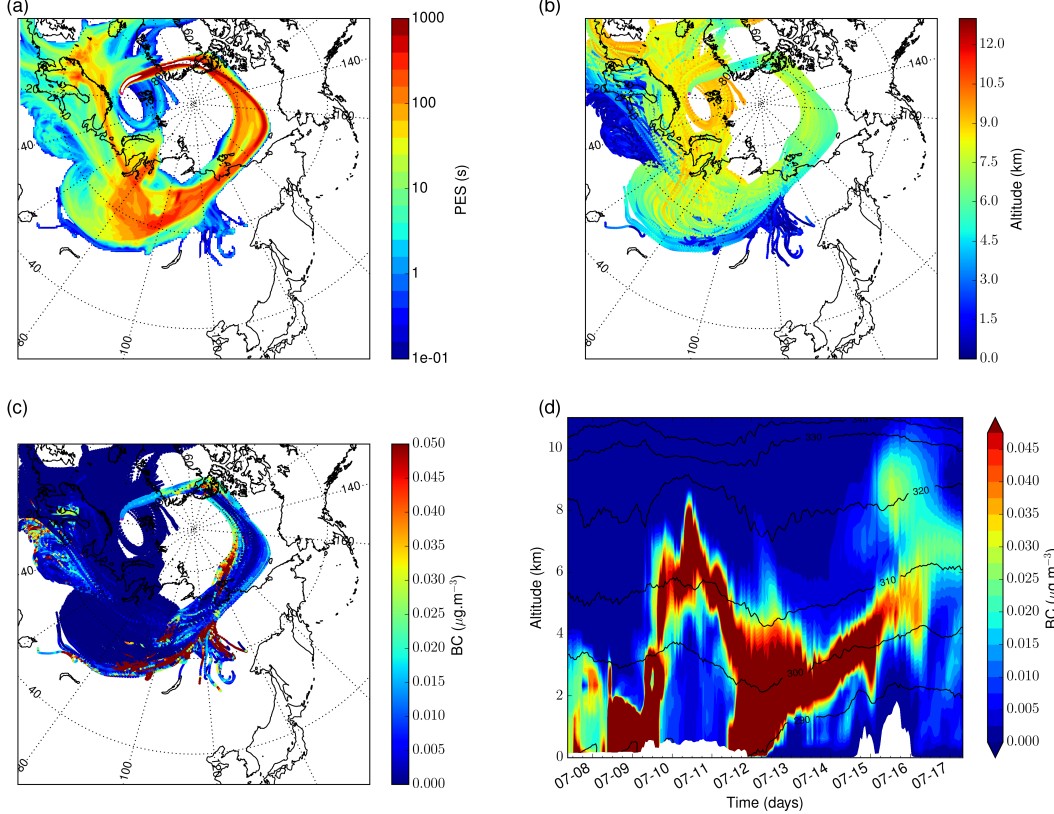

**Figure 10.** FLEXPART-WRF backward simulation from the biomass burning plume observed by the Falcon aircraft at $73.1°$N, $18.2°$E and 6.5 km between 08:37 and 08:47 UTC. (a): Column-integrated PES, (b) altitude and (c) BC mass concentration interpolated along the 10000 trajectories for 10 days prior to the release. In panel (d), the vertical cross-section of BC interpolated along the plume centroid location is shown, together with the dry isentropes (black lines) between 290 K and 340 K.

that the results are very similar for the four backward simulations in biomass burning plumes (not shown). Figure 10 highlights a cross-polar transport of biomass burning emissions from Siberian fires into the Arctic. They were probed in detail by the Falcon aircraft on the two flights on 17 July thanks to a pronounced northerly flow over the Svalbard archipelago. During the ACCESS campaign in July 2012, this transport of biomass burning pollution across the Pole was caused by Arctic low-pressure systems, regular phenomenons of the polar circulation especially during summer (Orsolini and Sorteberg, 2009). WRF-Chem meteorological analyses suggest that these Arctic transport events in the upper troposphere are due to WCBs associated with two wave cyclones over extreme northern Russia and eastern Siberia. The first one was present at $72°$N, $80°$E between 11 and 12 July and reached 970 hPa. The second cyclone, responsible of the plume curl over North Pole (Fig. 10), moved northward and extended from $80°$N, $140°$E to $88°$N, $180°$E, progressively deepening to reach a minimum surface pressure of 975 hPa.



Two large plumes enriched in CO and BC form over the south of the Krasnoyarsk region on 8-9 July and over Yakutia on 11-12 July (Fig. 7). Fire injection over those boreal sources facilitates a rapid uplift up to an altitude of 6 km. These plumes merge over eastern Siberia into a large plume advected to higher latitudes. When the first low-pressure system forms at the surface, the plume intrudes into the Arctic atmosphere, where it elongates and becomes narrower (12 July). The second low-

pressure system forms at 80°N, 140°E, which facilitates the progression of the eastern part of the plume further North and in altitude. Due to the progression of the cyclone to the North-East (88°N, 180°E), the plume elongates and moves almost over the North Pole (13 July). Along the plume boundaries, some mixing processes with the surrounding air occur, partly eroding its outer parts and leading to a filamentary structure. The ageing plume is split into two branches reaching northern Canada on 14 July and moving towards Svalbard. The relatively fine resolution of our simulations (40 km) can resolve the increased

filamentation of the pollution plumes, an advantage over most global models with coarser grids (Sodemann et al., 2011; Ma et al., 2014). The concentrations of CO and BC interpolated along trajectories generally decrease with time due to mixing with the surrounding clean atmosphere and deposition processes along transport (Fig. 10), except on 14-15 July when two plume branches of the same initial plume merge together.

Figure 9 also demonstrates a small contribution ( 15%) of anthropogenic sources to the enhanced CO and BC detected by the Falcon observations. As suggested by Fig. 10, the anthropogenic emissions that arrive near Norway may originate from fossil-fuel burning in northern China where high levels of BC have modeled observed 10 days before the flights on 17 July (Fig. 7). These emissions undergo a strong and efficient uplift from the PBL to the middle and upper troposphere within WCBs and therefore enter the Arctic at high altitudes. Those Asian anthropogenic air masses are characterized by high water vapor

mixing ratios ($14 \, \mathrm{g \, kg^{-1}}$) and RH values larger than 80%, facilitating their uplift over cold, dense Arctic air masses. The fact that the contribution of fossil-fuel emissions is found to be weak over Scandinavia suggests that significant deposition of aerosols occurs during transport. Wet removal is due to cloud formation and a high amount of precipitation during uplift, mostly convective rain (Fig. 5). It is also accompanied by an increase of potential temperature from 300 K to 315 K due to latent heat release from water vapor condensation.

According to Serreze and Barrett (2008) and Orsolini and Sorteberg (2009), such pollution transport events characterized by the presence of low-pressure systems in summer could be common and effective transport mechanism to the Arctic atmosphere for Siberian forest fires and Asian industrial emissions. Such transport pathways reaching the Arctic have been observed by Sodemann et al. (2011) and Roiger et al. (2011) during July 2008. In our study, the low-pressure systems are more intense

(surface pressures ranging from 970 to 975 hPa), suggesting an effective mechanism for intrusion of pollutants into northern Norway.

The pollution plumes exported from the flaring emission sources follow different pathways as a function of their altitude. The plume reaching Scandinavia in the PBL is exported above the West Siberian Plain where large values of the poleward

eddy heat flux is predicted (Fig. 7) and passes over the Barents Sea before progressing poleward at low level. On the contrary,





another plume emitted from the flares in western Siberia is transported eastward along the Siberian coast, moves to the North over Yakutia, curls over North Pole before reaching Svalbard in the mid-troposphere. Finally, the two anthropogenic plumes identified in the mid-troposphere close to Andøya island are exported from Europe in about 6 days.

## 5  Deposition during transport

### 5.1  Contribution of deposition processes

To understand the contributions of the different deposition processes (dry deposition, wet removal in grid-resolved clouds and in parameterized convective clouds) on the mass of BC transported to the region of the flights, we use the normalized differences between the NoDry, NoWet, NoWetCu simulations and the BASE run. Model results from those four runs are interpolated in time and latitude and longitude at the position of the Falcon during the two flights on 17 July. Figure 11 shows the time-altitude cross-sections of the relative contributions of the different deposition processes.

The dry deposition is only significant (more than $50\%$) in the lowest layers, and close to the Norwegian coast. Due to the fact that BC particles predominantly occupy the Aitken and accumulation mode ( $90\%$ of BC particles have a diameter lower than $2.5\,\mu m$ in our simulation), the impact of the sedimentation process is very weak. Dry deposition of BC-containing particles therefore arise from the aerodynamic transport down through the atmospheric layer to a very thin layer of stagnant air just adjacent to the surface (by turbulent diffusion) and from the Brownian transport across this quasi-laminar sublayer to the surface itself. As a consequence, the role of dry deposition is limited to the lower troposphere (PBL). More important are the impacts of wet deposition processes in grid-scale and subgrid parameterized clouds. Grid-scale wet removal has the largest effect ($80\%$ BC removal) in the mid-troposphere, where European influence is also large (Fig. 9). Altitudes influenced by biomass burning plumes (Fig. 9) experience similar BC removal rates, $40-60\%$, from both grid scale and subgrid scale clouds. In the areas where the biomass burning plumes were more intense, as identified in Fig. 4.2, the impact of the deposition processes is generally smaller, almost negligible for dry deposition, about $30\%$ for wet removal in grid scale clouds and $10\%$ in KFCuP. The fact that the maximum in BC concentrations correspond to zones less affected by deposition processes illustrates the heterogeneous vertical cross-sections of BC (Fig. 8). The vertical profile of BC (Fig. 3) would have therefore peaked more intensively in the mid- and upper troposphere if wet removal had been less efficient. Between $4$ and $6\,km$, the BC mixing ratio would have been multiplied by a factor of 3 without wet removal in subgrid convective clouds, or by a factor of 4 without wet removal in grid scale clouds. An understanding of the wet removal of BC is critically important because it directly controls vertical profiles of BC and amounts of BC transported from source regions to the Arctic.

BC particles coated with sufficient non-refractory and water-soluble secondary compounds are CCN (cloud condensation nuclei) active in liquid clouds. BC-containing particles can also act as IN (ice nuclei). In the MOSAIC aerosol module used in WRF-Chem, aerosol components are assumed to undergo a instantaneous internal mixing in each size bin. Petters et al.







**Figure 11.** Time-altitude cross-sections of the relative contributions (%) of (a) and (b) dry deposition, (c) and (d) wet deposition in grid-resolved clouds and (e) and (f) wet deposition in parameterized cumulus clouds to the BC mass mixing ratio along flight tracks 17a (in panels (a), (c) and (e)) and 17b (in panels (b), (d) and (f)). The altitude of the two flights is indicated in magenta in each panel.

(2009) showed that internally mixed BC particles can be more hygroscopic and easily removed by wet scavenging compared to





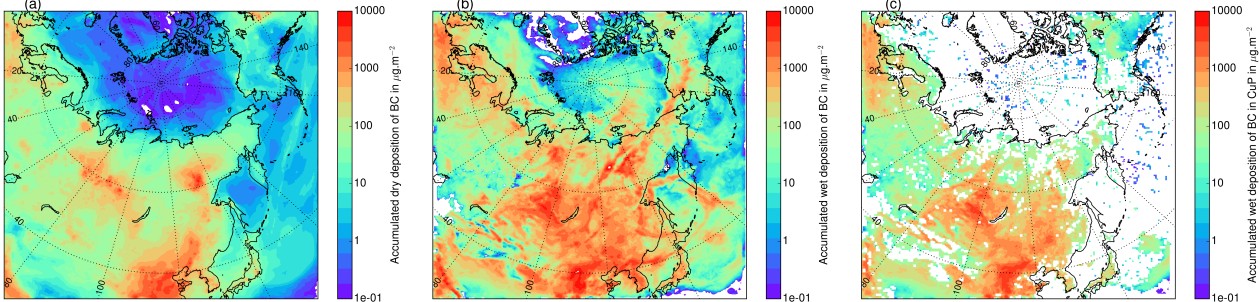

**Figure 12.** Accumulated deposition of BC due the (a) dry deposition, (b) wet removal in all clouds (grid-resolved and cumuli) and (c) wet removal in parameterized cumulus clouds between 7 and 21 July. Note the logarithmic color scale.

externally mixed BC. This should not have a large effect on our results as BC particles sampled by the Falcon have principally been transported in biomass burning plumes from Siberia (Sect. 4.4). Kondo et al. (2011) and Sahu et al. (2012) observed that the coating is often thicker in biomass burning plumes than in fossil fuel emissions, and is thought to occur in the first few hours after emission (Abel et al., 2003; Akagi et al., 2012).

During the long-range transport of plumes towards the Arctic region, the WCBs observed in this study loft aerosols from the PBL to the free troposphere, or rapidly inject them into the upper troposphere, in the warm section of mid-latitude cyclones. Clouds form and a large fraction of aerosols becoming efficient CCN is scavenged by cloud droplets or rain drops and potentially removed from the atmosphere via wet deposition associated with the cyclones. Aerosols that survive these vertical transport pathways to higher altitudes can travel to polar latitudes. But a small fraction of the remaining interstitial BC particles can be removed through impaction scavenging by collection of cloud or rain droplets. This process is said to be more effective for BC removal, both in mixed-phase (Twohy et al., 2010) and in ice clouds (Baumgardner et al., 2008; Stith et al., 2011). This is particularly important in this study as Fig. 6 indicated that ice, supercooled and mixed-phase clouds are predominant in the mid-troposphere during the ACCESS period.

## 5.2 Impact of deposition

Fig. 12 shows the accumulated deposition of BC between the 7 and the 21 July, which is the period during which aerosols were transported from mid-latitudes to the Scandinavia. The three types of deposition processes are studied here: dry deposition, wet removal by large-scale precipitation (first-order rainout and washout) and scavenging in wet convective updrafts (KFCuP). The deposition of BC is clearly dominated by wet removal from the middle to high latitudes. The impact of dry deposition can only been observed near emission sources: the spatial distribution of accumulated BC by dry deposition is co-located with the map of the upward BC flux and the patterns of the divergence of horizontal BC flux in PBL (Fig.7). It is therefore similar to the spatial distribution of emissions. Sedimentation and turbulent mix-out are indeed the dominant sink of large particles contain-





ing BC near emission sources. The dry deposition flux of BC particles from the troposphere to the surface is moreover almost proportional to the local BC mass mixing ratio near the surface. In large scale clouds, BC particles are removed through the nucleation scavenging mechanism or through below-cloud scavenging, where particles are collected by falling hydrometeors (in stratiform precipitation) through Brownian motion, electrostatic forces, collision or impaction (Seinfeld and Pandis, 2006).

Interstitial and cloud-borne BC-containing particles can finally be removed in the strong updraft core of deep convective clouds parameterized in KFCuP.

The horizontal distribution of BC wet removal results from a combined effect of the precipitation amounts (Fig. 5) and the upward BC flux (Fig. 7). During the vertical transport of plumes containing carbonaceous aerosols, the wet deposition is indeed

directly linked to the distribution of precipitation. The spatial distribution of BC scavenged by wet deposition in parameterized cumulus clouds matches well that of convective precipitation (Fig. 5). Aerosol wet removal from predicted deep and shallow convective clouds is however lower than the removal from grid scale clouds. This is the case even where the cumulus scheme predicts greater amounts of precipitation than the microphysical scheme (Fig. 5). KFCuP triggers almost everywhere in the mid-latitudes, but not so much over the oceans and in the Arctic or sub-Arctic (beyond $60°$N) where it is mostly dominated by

the formation of low-level stratocumulus cloud decks in the boundary layer and lower troposphere, producing frequent drizzle (Browse et al., 2012). We have to note that the KFCuP mechanisms actually lacks below-cloud wet scavenging and scavenging by snow and ice. Adding the below-cloud scavenging by the convective precipitation would nevertheless have only a minor impact on the BC concentrations: assuming that nearly all the BC was in the $80-470$ nm diameter range, the below-cloud removal efficiencies are indeed very small. Some of the enhanced wet removal in cumuli must be compensated by the en-

hanced vertical transport of aerosols from the PBL into the free troposphere during deep convection (Berg et al., 2015). It is also important to bear in mind that the cumulus cloud fraction within the model grid box (here $40$ km) can be quite small, so that the wet removal occurs over a relatively small area and does not have a large impact on the total aerosol loading within the model grid box. While validating the KFCuP cumulus scheme, Berg et al. (2015) highlighted a reduction of low-altitude aerosol associated with the venting of aerosol from the PBL as well as changes in the vertical structure associated with the

cumulus induced subsidence, but little change in the average aerosol aloft. This may be also consistent with the small amount of secondary activation in the simulations.

The fact that wet deposition is predominant at all latitudes during summer 2012 suggests that a large portion of BC is removed in the regions south of $70°$N before reaching the Arctic. Matsui et al. (2011) and Sharma et al. (2013) confirmed that

the BC emissions in the East-Asia region ($30-40°$N) are subject to more efficient scavenging ($90\%$ wet deposition) than other temperate source regions, meaning the East-Asian influence on the Arctic is significantly inhibited compared to other regions. Due to stronger scavenging in East-Asia, this source region has less influence on the Arctic compared to biomass burning sources in Russia (Sect. 4.4), where precipitation rates are lower. The influence of wet removal of BC scavenging is also predicted in the high latitudes in summer, underlining the role of drizzle from low-level clouds and fogs. This scavenging

at low level inside the Arctic region associated with a decrease in transport from outside the Arctic are responsible from the low





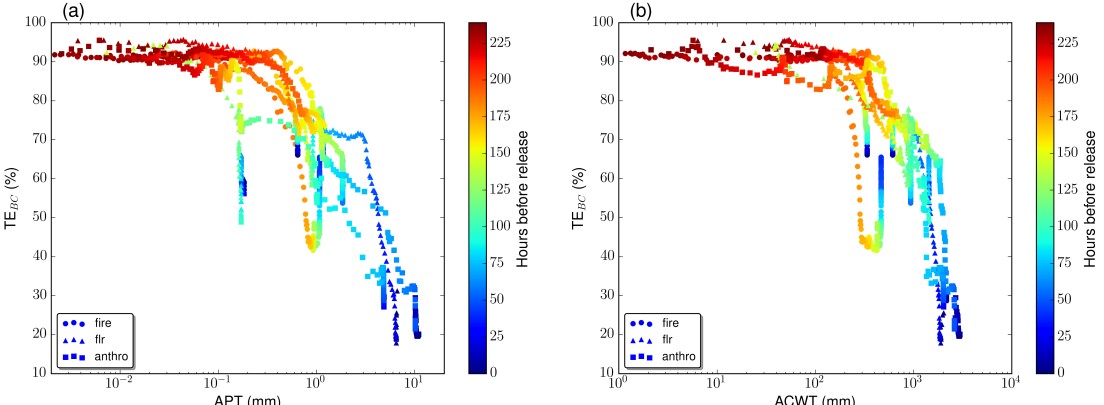

**Figure 13.** Transport efficiency of BC ($TE_{BC}$) for eight plumes identified from sampled air parcels by the Falcon (4 boreal fire plumes, 2 plumes from flaring activities in Northern Russia and 2 anthropogenic plumes from Europe) as a function of (a) the accumulated precipitation along trajectory (APT) and (b) the accumulated condensed water along trajectory. The parameter is the time (in hours) before the release of the trajectories in FLEXPART-WRF (color scale).

summertime BC concentrations. We note the fact that the WRF-Chem performs worse at $100$ km to reproduce the mean BC profile observed by the Falcon particularly in the mid-troposphere (Fig. 3) suggests that the coarse resolution and the increased role of the KFCuP mechanism result in less efficient BC removal during transport. This is due to the presence of weakly precipitating shallow clouds at high latitudes, that do not scavenge aerosols adequately.

## 5.3 Transport efficiency of BC particles

### 5.3.1 Role of accumulated precipitation and condensed water along FLEXPART-WRF trajectories

To investigate the transport efficiency of BC particles, we performed a simulation that did not consider aerosol wet deposition both in grid scale and in parameterized convective clouds (NoWetAll run), and the results of this simulation are compared with

10 those of the BASE simulation to evaluate the effects of the wet removal of BC from the atmosphere by precipitation during transport. We use CO as an inert combustion tracer (lifetime of a few weeks) that is not generally impacted by precipitation during transport from Siberian and East Asia. We define the transport efficiency of BC as a function of altitude $z$ as:

$$TE_{BC}(z) = \frac{\left(\frac{\Delta[BC](z)}{\Delta[CO](z)}\right)_{BASE}}{\left(\frac{\Delta[BC](z)}{\Delta[CO](z)}\right)_{NoWetAll}} \tag{1}$$

where $\Delta[BC]$ and $\Delta[CO]$ represent the differences between simulated and background values of BC mass mixing ratios, re-

15 spectively. Similar quantities have been used in previous studies (e.g., Koike et al., 2003; Matsui et al., 2011; Oshima et al.,



2012), but the denominator was often chosen as the BC-to-CO ratios for the source regions rather than its value extracted from a simulation where wet removal has been turned off, as in Oshima et al. (2013). In the latter study however, the modeled $TE_{BC}$ values did not take into account the CO concentrations. Our method has the advantage that the depletion of the BC-to-CO ratios is only due to wet removal (both in grid scale and parameterized clouds), and not to the fact that some plumes emitted from the source regions do not finally reach the receptor area (Scandinavia) due to different transport patterns or diffusion during transport. The background value of CO has been determined at each height by the fifth percentile of all CO measurements (Fig. 3) in an altitude range 500 m-thick centered around that height. The background value of BC has been assumed to be zero, due to its short lifetime (less than one week).

To evaluate the aerosol wet removal processes used in the model in more detail, we examined the dependence of $TE_{BC}$ on the amount of precipitation that an air parcel had experienced during transport. Following the methodology of Oshima et al. (2012) who investigated the role of wet removal of BC in Asian outflow towards the Pacific Ocean, we estimate the accumulated precipitation along trajectory (APT) and use it as a proxy for wet deposition. For every sampled air parcel by the Falcon and identified as a plume on 17 July in Sect. 4.4, the $TE_{BC}$ and APT values are computed along each of the 10000 FLEXPART-WRF trajectories released at the time and location of the plume. Specifically, the APT is derived by integrating the WRF-Chem precipitation amounts from the BASE run along each Lagrangian 10 days backward trajectory of the corresponding uplifted air parcel. For the plume originating from Russian flaring sources (rapidly transported at a low altitude (Sect. 4.4)), the trajectory runs backwards for 6 days. At hourly intervals, the rain, ice, snow and graupel precipitation rates (including values in cumuli) are interpolated at the closest WRF-Chem grid box of the FLEXPART-WRF trajectory, summed and then integrated in time. This method is preferred to the one based on the WRF-Chem surface precipitation amounts that would not take into account the relative vertical distribution of precipitation occurrence and air parcels. Similarly, we also extract the accumulated condensed water along trajectory (ACWT) to account for the fact that BC particles are also removed from the plumes through the nucleation scavenging mechanism. The ACWT is calculated using the method applied for APT but based on the mixing ratios of cloud liquid water, ice, snow, rain and graupel contents both in grid scale and convective parameterized clouds. The mixing ratios are summed and integrated vertically in each model grid box crossed by the trajectory and finally along this trajectory. $TE_{BC}$, APT and ACWT values extracted along each of the trajectories are then averaged to give a mean trajectory corresponding to an uplifted air parcel. As presented in Sect. 4.4, this method is applied to 4 boreal fire plumes, 2 plumes from flaring activities in Northern Russia and 2 anthropogenic plumes from Europe.

### 5.3.2 Results along trajectories

In Sect. 4.4, eight distinct plumes have been identified: 4 plumes originating from boreal fire sources in Russia (associated with a small influence of anthropogenic Asian air masses (Fig 9)), 2 plumes from flaring activities in Northern Russia and 2 anthropogenic plumes from Europe. Figure 13 shows the $TE_{BC}$ values for these eight plumes as a function of the APT, ACWT and FLEXPART-WRF transport time. $TE_{BC}$ systematically decreases with increasing APT, reflecting the crucial role of precip-



itation in removing particles containing BC along transport, and thereby in controlling the amount of BC reaching the Arctic. A similar behavior is observed plotting $TE_{BC}$ as a function of ACWT, suggesting that BC can also be removed efficiently by nucleation scavenging when transported to the Arctic. An exception occurs on 14-15 July, when $TE_{BC}$ slightly increases in some plumes, because two branches of the same initial plume merge over Northern Canada (Sect. 4.4). $TE_{BC}$ decline is not

linear with APT or ACWT (logarithmic scale in Fig. 13) but starts to significantly decrease $6 - 7$ days ($144 - 168$ hours) before reaching Scandinavia, corresponding to APT values of $0.2 - 0.4$ mm and ACWT between $200$ and $400$ mm.

When reaching Scandinavia (retroplume age very small), plumes present various values of $TE_{BC}$ as a function of their origins. $TE_{BC}$ is the greatest for Siberian biomass burning impacted air masses. As shown in Fig. 9, $85\%$ of the BC sampled by

the aircraft in such plumes originated from boreal fires and $15\%$ from Asian anthropogenic masses. In such plumes, the $TE_{BC}$ was as high as $56 - 68\%$ and was caused by low APT and ACWT values ($1$ mm and $300 - 1000$ mm, respectively). In contrast, $TE_{BC}$ was low ($21 - 28\%$) and APT and ACWT amounts were high for European anthropogenic air parcels ($5 - 10$ mm and $2000 - 3000$ mm, respectively). Air parcels influenced by flaring emissions exhibit very different behavior as a function of their altitude: moderate $TE_{BC}$ values ($50\%$) were calculated in plumes reaching the middle Arctic troposphere ($3.7$ km), whereas

$TE_{BC}$ decreased to $20\%$ in plumes transported in the PBL. ACWT appears to be a better proxy for characterizing the efficiency of deposition processes in such air parcels, since the values are of the same order of magnitude in both flaring impacted air masses ($1000 - 2000$ mm), while APT varies from $0.2$ mm (in mid-tropospheric plumes) to $8$ mm (in PBL plumes).

The magnitude of the differences in $TE_{BC}$ values obtained during the same period due to various origins is particularly

noticeable. European anthropogenic air parcels and flaring emissions are generally transported at lower altitudes than biomass burning air masses, according to Fig. 8. They encountered more (non-convective) precipitation caused by frequent drizzle in low-level stratocumulus clouds, scavenging aerosols during transport. In contrast, Russian and to a lesser extent Asian air masses have exported BC more efficiently towards the Arctic. Most air parcels from those sources have experienced transport in WCBs, removing part of the BC loadings. But, the transport of aerosols has been facilitated thanks to the injection of fire

plumes at high altitudes and to the fact that the locations of biomass burning in Siberia are very septentrional sources in summer ($45 - 65°$N).

### 5.3.3 Zonal mean of transport efficiency

The vertical distributions of the mean BC mass concentrations zonally averaged during the ACCESS period in the BASE run

and the corresponding mean $TE_{BC}$ values due to grid scale clouds and all clouds are shown in Fig. 14. The BC mass concentrations are greatest below the altitude of $4$ km on the mid-latitude region ($40 - 66°$N) with values larger than $50$ ng m$^{-3}$. The highest concentrations ($> 100$ ng m$^{-3}$) are found close to emission sources and they continuously decrease with height. A high contrast is observed between the mid-latitudes and the Arctic. In the mid-troposphere ($3 - 6$ km), the contrast is less pronounced as moderate BC concentrations are spread on a wide latitude range. As a consequence, above $80°$N, lowest BC



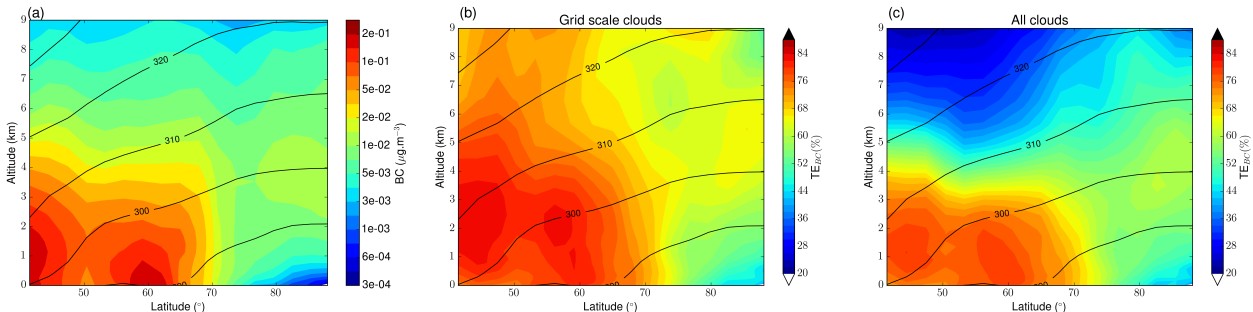

**Figure 14.** Latitude-altitude cross-sections of (a) the BC mass concentrations, and the corresponding TE$_{BC}$ values due to (b) grid scale clouds and (c) all clouds, averaged zonally and temporally over the ACCESS campaign period in the BASE run. On each panel, black solid lines represent the dry isentropes between 290 K and 330 K.

concentrations are found at the surface, and the maximum is detected between 2 and 4 km (15 ng m$^{-3}$).

In contrast, TE$_{BC}$ due to only grid scale precipitation exhibits a very distinct dependency on latitude: above 70°N, TE$_{BC}$ decreases from $70-85\%$ to $44-66\%$, illustrating the sharp meridional gradient in the distribution of moisture and precipitation.

This explains the contrast between the BC concentrations in the mid-latitudes and in the Arctic region. Within the Arctic, the maximum of TE$_{BC}$ ($\simeq 66\%$) is found at higher altitudes (4 and 6 km) than the maximum in BC concentrations. This provides an evidence that the transport in WCBs partly, but not completely, removes carbonaceous aerosols by wet deposition. The low BC concentrations in the lowest polar layers are correlated with a weak transport efficiency ( 45%). This is due to frequent drizzle produced by the formation of low-level stratocumulus clouds in the Arctic PBL and lower troposphere, together with

a slower transport in the lowest altitudes. Is is also interesting to note that in the mid-latitude regions, higher TE$_{BC}$ are decoupled from the surface, reflecting the strong uplift of Siberian fires emissions above boreal forest. The associated strong ascent generally occurs at subgrid scale ($< 40$ km), which explains the inability of wet removal in large scale precipitation to clean the atmosphere above sources.

TE$_{BC}$ due to wet removal in all clouds exhibits a totally different pattern. TE$_{BC}$ values are correlated more closely with BC mass concentrations, suggesting that wet removal in subgrid parameterized clouds (cumuli) is also fundamental to understand the vertical distribution of BC mass. It also suggests that, in cumuli, the effect of wet deposition in the mid-latitudes is more important than the enhanced vertical transport in reducing low-level aerosols. The interactions between aerosols and clouds in cumulus clouds is responsible of the vertical gradient of TE$_{BC}$, especially in the mid-latitudes. Two noticeable differences

between the zonal cross-sections of BC and TE$_{BC}$ are observed. First, in mid-latitudes ($40-70°$N), TE$_{BC}$ is very low above 6 km (TE$_{BC} < 20\%$), whereas BC concentrations reach moderate values in this altitude range: this reflects the very efficient BC wet removal due to convection and precipitation during the rapid ascent over the source regions. The BC mass concentrations observed above 6 km mainly arise from the boundaries. Second, in the Arctic, the maximum of TE$_{BC}$ is located between 3.5





and 5.5 km (as high as 64%): it illustrates the fact that the transport of Siberian fires plumes closely followed isotropes, as also noticed by Matsui et al. (2011). Aerosol removal strongly modulates long-range transport from Asia and Siberia to the Arctic, with a strong zonal component and transport at higher elevations. It is important to note that the transport efficiency is probably a bit overestimated in our study because the model slightly over predicts BC mixing ratios in the mid-troposphere (Fig. 3).

The Siberian plume has been sampled by the Falcon in the upper troposphere ($6 - 8$ km). In Fig. 14, we find that $TE_{BC}$ is about 60%, which is in agreement with the value found by along trajectories (Fig. 13). The altitude dependency of the BC mass concentrations in the ACCESS aircraft measurements (Fig. 3 and Fig. 8) is consistent with the zonally and temporally averaged $TE_{BC}$ distribution simulated in the BASE simulation during the ACCESS campaign period. This suggests that the

general features of wet removal of BC transported in plumes from Siberia and Asia during summer 2012 were well captured in the ACCESS aircraft measurements. We finally note that the $TE_{BC}$ values in this study are much higher than in those reported by Matsui et al. (2011). That discrepancy is partially due to the different year (2008 vs 2012) with different emissions and precipitation patterns (APT values much lower in our study). In addition, the definition of $TE_{BC}$, used by Matsui et al. (2011), is the transport efficiency of BC uplifted air masses sampled by the aircraft relatively to their corresponding sources. In our study,

$TE_{BC}$ represents the transport efficiency of BC due only to wet removal in grid scale and subgrid parameterized convective clouds. The fact that some plumes are exported from the same emission source to different destinations (e.g. over the Pacific Ocean, as in Oshima et al. (2012)) does not influence our results.

## 6 Summary and conclusions

During the ACCESS campaign in summer 2012, extensive boreal forest fires associated with temperatures record in both western and eastern Siberia strongly impacted the atmospheric composition of the Arctic atmosphere. The Falcon aircraft regularly sampled transported pollution. During two dedicated flights on 17 July, it measured plumes in the middle and upper troposphere that were representative of cross-polar transport from Siberia and Asia towards Scandinavia and Svalbard archipelago. Specifically, enhanced CO concentrations and BC mixing ratios up to 200 ppbv and 25 ng kg$^{-1}$ are reported between 7 and 8

km altitude, reflecting the influence of pollution transported from remote sources. The WRF-Chem model run at a horizontal resolution of 40 km on a polar stereographic projection shows good skill in capturing the variability seen in the vertical distribution of chemical profiles, with $NMB$ of 1.5% and 27.3% for CO and BC respectively. BC increases in the middle and upper troposphere are mostly linked to biomass burning plumes transported from boreal sources (85%), weakly influenced by anthropogenic sources in northern China (15%).

Our results underline that a coarser horizontal resolution (100 km instead of 40 km) deteriorates the model performance, with a significant overestimation of BC levels in the mid-tropohere. This suggests that, at a coarser resolution, the model is unable to resolve the fine structure of plumes transported in altitude across the North Pole, illustrates the difficulty to represent



the cloud and precipitation structures and scavenging processes at subgrid scale and points for the need for improved representation of BC processing in global models in the Arctic.

With a horizontal resolution of $40$ km, a very good spatial correlation is found between the horizontal distribution of AOD, the strong divergence regions of the BC horizontal fluxes in the PBL, and the spatial distribution of the mean upward BC mass fluxes, underlying ideal conditions for BC poleward transport. This highlights two major biomass burning regions in Siberia, south of Krasnoyarsk region and Yakutia, and one source of anthropogenic pollution in central and northeastern China. Aerosols and CO emitted from the two boreal forest fires regions are exported towards the Arctic through the North of Yakutia in the region presenting a large horizontal poleward eddy heat flux. Their transport towards the North Pole is caused by low-pressure systems controlling the progression and path of the plumes. The plume-rise injection model embedded into WRF-Chem high resolution simulation correctly represents the mixing processes at the edge of the plumes as well as plume filamentation. In the lowest Arctic atmospheric layers, weak CO and BC enhancements are linked to the inefficient transport of European anthropogenic emissions and to a lesser extent to emissions from flaring activities in Northern Russia. These emissions are exported over the Barents Sea and transported at low altitudes into the Arctic.

During the long-range transport of Siberian and Asian plumes to the Arctic region, a large fraction of BC particles coated with sufficient water-soluble compounds become CCN active and are scavenged by cloud droplets or rain drops via wet deposition associated with the warm section of mid-latitude cyclones. The two wet deposition processes, namely the wet removal by large-scale precipitation and the scavenging in wet convective updrafts contribute almost similarly to the total accumulated deposition of BC. The weak role of dry deposition is limited to the lower troposphere. The simulated transport efficiency of BC ($TE_{BC}$), accumulated precipitation along trajectory (APT) and accumulated condensed water along trajectory (ACWT) values provide insights for understanding the wet removal of aerosols transported to the Arctic. The $TE_{BC}$ in biomass burning plumes is about $60\%$ and is found to be due to low APT ($1$ mm). In contrast, $TE_{BC}$ is very small ($< 30\%$) and APT are larger ($5-10$ mm) in plumes influenced by urban anthropogenic sources or flaring activities in Northern Russia and transported at lower altitudes.

The transport efficiency of BC due to grid scale precipitation is responsible for the sharp meridional gradient in the distribution of BC concentrations. The transport in WCBs, rapidly injecting aerosols into the upper troposphere, removes some but not all carbonaceous aerosols by wet deposition. The wet removal from subgrid parameterized clouds (cumuli) in the mid-latitudes is found to be more important than the enhanced vertical transport in reducing low-level aerosols and is responsible for the vertical gradient of $TE_{BC}$, especially in the mid-latitudes. Wet deposition in parameterized subgrid clouds is essentially active below $60°N$, reflecting the distribution of convective precipitation, but is dominated in the Arctic region by the grid-scale wet removal associated with the formation of stratocumulus clouds in the PBL producing frequent drizzle. An understanding of the respective contributions of the different wet removal processes of BC is therefore critically important because it drives the vertical profiles of BC and strongly modulates the long-range transport of BC from Asia and Siberia to the Arctic, with a strong





zonal component and transport at higher elevations. The general features of wet removal of BC transported in plumes from Siberia and Asia during summer 2012 were well captured in the ACCESS aircraft measurements.

*Author contributions.* The first author performed the WRF-Chem simulations, the analyses and drafted the paper ; the co-authors contributed text, ideas and discussed the results. Most of the authors participated in the ACCESS campaign in July 2012.

5 *Acknowledgements.* The research leading to these results received funding from the European Union under Grant Agreement 265863-ACCESS (www.access-eu.org)-within the Ocean of Tomorrow call of the European Commission Seventh Framework Programme. French co-authors acknowledge support from the national Chantier Arctique project, PARCS (Pollution in the Arctic System), funded by CNRS. We thank the pilots, engineers, and scientists from the DLR flight department for their excellent support during the field campaign. Computer modeling performed by J.-C. Raut benefited from access to IDRIS HPC resources (GENCI allocations 2015-017141 and 2016-017141) and
10 the IPSL mesoscale computing center (CICLAD: Calcul Intensif pour le Climat, l'Atmosphère et la Dynamique). We thank the EDGAR team for compiling the HTAPv2 emissions. J.D. Fast and L.K. Berg were supported by the Office of Science of the U.S. Department of Energy as part of the Atmospheric System Research (ASR) program. The Pacific Northwest National Laboratory (PNNL) is operated by DOE by the Battelle Memorial Institute under contract DE-A06-76RLO976 1830.





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
