# Peer review of "Cross-polar transport and scavenging of Siberian aerosols containing black carbon during the 2012 ACCESS summer campaign"

_Atmospheric Chemistry and Physics, 2016_

## Referee Comment (RC1) · Anonymous Referee #1 · 17 Feb 2017

Review of "Cross-polar transport and scavenging of Siberian aerosols containing black carbon during the 2012 ACCESS summer campaign" by Raut et al.

**General comments**

This paper presents a thorough and detailed model investigation of the factors affecting transport of several plumes of BC&CO to the Arctic, comparing the influences of different pollution sources and the spatial distribution, amount and types of precipitation. There is a limited comparison to aircraft measurements that shows the model is doing a reasonable job of capturing the general features of the observations, though not doing a perfect job of simulating the exact concentrations or resolving the finer structure of the plumes (and I would not expect it to). Overall I think the work is of a very high scientific quality, and the main issue I have is that it is too long. However, I think it is probably not suitable to be split into 2 papers, nor are there any sections that could easily be cut without detracting from the rigour of the analysis, so I think it will just have to remain long. Possibly, parts of Section 2 might work as an appendix/supplement. Additionally, some of the written English is phrased in strange ways and some sentences do not quite make sense, so it could do with some work. I recommend publication in ACP subject to the following minor revisions.

**Specific comments**

P1L13-15 The way this is phrased makes it sound like the source determines the APT. I think this is because you have said "is" rather than "was", which makes it sound more like a general statement rather than a specific statement about the plumes studied in this paper. This type of error occurs throughout the paper and is mostly benign but can sometimes be confusing, such as in the example above. As a general rule of thumb, the work you have done is in the past (e.g. "the campaign took place" or "we ran the model") but things you do in the paper itself should be in the present (e.g. "in this study we describe…" or "Figure 5 shows …". It gets more tricky, for example when you come to the conclusions- the specific plumes you studied *were* affected by precipitation but plumes in general *are* affected by precipitation.

P3L8-9 "Schwarz et al showed…." I'm not sure what you are trying to say with this sentence, other than to mention the study by Schwarz et al. It seems like you do a better job of saying the same thing in the next sentence, so this one isn't really necessary

P3L15 What results from Koch and Hansen? Do they confirm them or do they just agree with them? (i.e. both could be wrong)

P5L6-10 Please give details of how the SP2 was calibrated. Actually, you should also say how the CO box was calibrated. Just one sentence for each would probably be sufficient if a standard method was used.

P5L14-15 "Absolute uncertainty of BC particle mass is within 10%, the uncertainty of the derived total BC mass mixing ratio is about 30%." I am not sure I follow the logic here. Also 10% is not an absolute uncertainty, it is a relative uncertainty- a percentage is relative by definition. Given you do not mention particle size in your analysis, the only relevant errors are A) The systematic uncertainty in your BC calibration and B) The statistical uncertainty in the derived BC mass concentrations.

A) is down to a combination of the sampling time, concentrations and flowrate, but is easy as you can just pick a time when you think your concentrations are constant and look at the variation in your time series. You could express this as a relative error (e.g. +/- 10%) or absolute (e.g. +/- 2 ng/kg)

B) is more difficult as you have two factors- firstly the random variation in your calibration slope (in other words how accurately does your particular slope recreate the mass of the calibration material, if you repeated your calibration exactly how much would the slopes differ?) (see Laborde et al. (2012b)) and secondly how well does your calibration material represent the instrument response to the BC you measure in the atmosphere? As Laborde et al. (2012a) showed, the SP2 responds differently to different BC types. I understand the observations present a fairly minor part of your paper but if you are going to present them and quote an uncertainty it should be done correctly, which at present I don't think it is.

P7L23 You note that the height of the emissions injection is very important- how good a job does the plume rise model do? How does it work out the buoyancy of a particular fire? Please provide a brief summary

Figure 2a is there a reason why potential temperature is more useful than just temperature?

P10L20 wrong OH and transport- the way this is written it is not quite clear what you mean. Wrong transport? Vertical or horizontal? Or just transport? Do these factors and the studies you reference explain why CO is underestimated specifically between 6 – 9km?

Section 3.2 I think you are overselling the agreement between model and measurements. For example "The two profiles are well correlated with maximum CO values of 200 ppbv at 7–8 km, associated with elevated BC values reaching 25 ng kg−1." Actually the maximum in the model CO is ~150ppb at 6.5km. Compare to P32L10- this is a better way of describing the model/measurement comparison. The general features were well captured.

P11 L 1 "the influence of flaring emissions in this area is insignificant". Insignificant for what? At what scale? Later on in the paper you talk about some flaring plumes so it can't have been insignificant within those plumes.

P11L7-8 "The model shows appreciable skill in capturing the vertical profile of BC, but overestimates the BC mixing ratio between 2 and 3 km of altitude." Looking at the median BC concentrations, the 40km model overestimates between 1.5 – 7.5km then underestimates higher than 7.5km. Additionally, if you just went with a flat BC concentration of around 6 or 7 ng/kg the medians would probably show similar agreement. Now, I am not saying that the agreement is terrible- actually it OK, perhaps reasonably good, and it seems to capture the general features of the observations without doing a perfect job. But you read the text and it sounds like the model is doing an amazingly good job, which figures 3 and 8 show it isn't, it's just doing a reasonable job. So I think you should just tone down how well you claim the model and observations agree.

P11L9 The 30% error in the SP2- please relate this back to the previous comment on statistical vs systematic error. Also please give numbers for the model biases.

P11L13 Here you say the CO between 6-9km is due to biomass burning emissions, but previously you said the underestimation in CO was due to wrong OH concentrations and transport. Could it not be

that the model is underestimating the biomass burning CO at this altitude? Is that what you mean by transport? Please clarify

P13L11 You say the AOD underestimation is due to simplified SOA, but you haven't given any details of how the AOD is calculated. I don't think you can say this is due to missing SOA when you don't have a good handle on even the size distribution, let alone composition or refractive index.

P18L13 are the BC enhancements you are talking about in the model or in the observations? I assume model but it's not clear

P18L23 The agreement between model and measured BC is "very good"- Again I would say it's reasonable but I wouldn't say it's "very good". The model plumes in figure 5 are too diffuse and some are missing, such as the smaller amounts of BC associated with the CO plume at ~8km at 0915. You get the general features. For me, "very good" would be if you could plot model vs measurement for each grid box as the aircraft passed through it and get something approaching a 1:1 line, though I doubt that could happen in a study like this.

Figure 8 You could do with a longer time average of the measurement data as currently it's difficult to see the structure when the markers all overlap. I also wonder if there is a way to make the aircraft easier to see as the parts that stand out are the parts where it disagrees with the model. Also it might help if the x-axis was north/south or east/west as several points in the discussion relate to spatial location and this is difficult to see in this and subsequent plots

P21L18 If the aerosol from flaring had been removed by precipitation, wouldn't you still see the CO?

P21L25 Didn't you say in the previous section that the flaring plumes didn't exist?

Section 4.4 The discussion may be easier to follow here if you circled the plumes on one of the figures

P23L2 you say the fire injection lofted the plumes to 6km, but doesn't fig 10d shows that the emissions from the 8[th] initially remained below 4km. I'm not really sure I follow what figure 10d adds to the analysis

P24L20 How does figure 9 show European influence?

P24L24 Please define KFCuP in the text. You refer several times to KFCuP as if it is a process itself. In the real atmosphere it is actually "convective clouds" that do things that the model is trying to represent.

P26L4 It's frowned upon these days to refer to it as a coating, as that may not represent the actual morphology of the particle. Maybe say BC in BB plumes is more internally mixed or something like that. I also saw another point where you referred to coatings, please also change this.

P26L11-14 I don't know what these last 2 sentences add to the analysis

P27L19 The removal efficiencies may be low for large rain drops but not for drizzle

P29L6 The 5[th] percentile of the measured or the modelled CO concentration? Also why not do this for the BC as well? The average lifetime of aerosols is of the order of a week because of deposition

processes, mostly wet deposition. So the lifetime of aerosol that escapes wet deposition is longer. If you mean you looked at figure 3 and saw that the minimum BC in the observations was basically zero at all levels then say that. But if that was the case I still don't see the harm in taking the 5[th] percentile like you do with the CO.

P29L22 Perhaps this is not the case in the model but in the real atmosphere BC would only be lost if the cloud precipitated. If it evaporated the BC would still be there.

P32L16 Can you suggest how the discrepancy/difference might be resolved? Is it just because there is less precipitation in this study?

P33L30 "found to be more important" It is not clear what you mean by this. If you mean the cumulus clouds remove more of the low-level aerosol by scavenging or washout than by uplifting it then say that. It's better to say what is actually happening.

P34L1 This last sentence is an odd way to end a paper. It sounds like you are using the model to validate the measurements when actually it was the other way round. Also this wasn't the main focus of your analysis. It seems to me that the main important conclusions were

1) BC is transported more efficiently into arctic from high-latitude BB sources than from east-Asian anthropogenic sources because it rains less at higher latitudes

2) The ways in which the large-scale vs subgrid convective clouds affected the BC distributions differently.

It would be good if you could highlight these more in the abstract/conclusions in terms of the physical processes your results suggest are actually taking place in the real atmosphere, rather than abstract terms like grid-scale and APT

Finally, given you have actual observations it seems like there is a missed opportunity to calculate $TE_{BC}$ based on the measured values of BC and CO. You have hinted at your reasons for not doing so in the comparison to previous studies calculating $TE_{BC}$ but I'm not sure it's clear exactly why not.

**Technical corrections**

P3L25 "lead to in the" remove 'in'

P3L27-31 This is a very long sentence, please split it into at least 2.

P8L4 add "resolution" before "to adequately"

P11L12 remove "but"

Figure 3 please make the plots slightly taller to show the detail better

P11L19 This is a very long sentence, please split it into at least 2.

P12 L4 remove "that"

Figure 4 the scale is between 0 – 1 but the text mentions values up to 2.5

Figure 5 are (a) and (b) really the same grid? Panel (a) looks much more blocky. Also label (%) on panel (c) colorscale

P15L1 "than in altitude", do you mean "that at higher altitudes"?

P18L6 "plumes transported in altitude" do you mean "at altitude"?

P18L20 "shifted a bit"- how far? Saying "a bit" is probably *a bit* too colloquial for a paper (but not a review!)

P21L17 "not due to" do you mean "did not lead to"?

P27L34-35 This sentence doesn't make sense

Figure 13 caption Maybe rephrase to "The points are colored by the time (in hours) before the release of the trajectories in FLEXPART-WRF)"

P30L25 You may consider not using the word "septenrional" as it is not a commonly used word.

P32L4 Again change "a bit" to "slightly"

P33L4 Remove "very"

Finally, thankyou for your interesting (but long!) paper

**References**

Laborde, M., Mertes, P., Zieger, P., Dommen, J., Baltensperger, U. and Gysel, M.: Sensitivity of the Single Particle Soot Photometer to different black carbon types, Atmos. Meas. Tech., 5(5), 1031–1043, doi:10.5194/amt-5-1031-2012, 2012a.

Laborde, M., Schnaiter, M., Linke, C., Saathoff, H., Naumann, K.-H., Möhler, O., Berlenz, S., Wagner, U., Taylor, J. W., Liu, D., Flynn, M., Allan, J. D., Coe, H., Heimerl, K., Dahlkötter, F., Weinzierl, B., Wollny, A. G., Zanatta, M., Cozic, J., Laj, P., Hitzenberger, R., Schwarz, J. P. and Gysel, M.: Single Particle Soot Photometer intercomparison at the AIDA chamber, Atmos. Meas. Tech., 5(12), 3077–3097, doi:10.5194/amt-5-3077-2012, 2012b.

---

## Referee Comment (RC2) · Anonymous Referee #2 · 24 Mar 2017

**Review of "Cross-polar transport and scavenging of Siberian aerosols containing black carbon during the 2012 ACCESS summer campaign" by J.-C. Raut et al.**

**General comments:**

This work investigates the transport processes of black carbon (BC) from Siberia to the Arctic and the effects of wet removal over mid- and high-latitudes during the ACCESS aircraft campaign using a regional scale chemical transport model (WRF-Chem) and a Lagrangian particle dispersion model (FLEXPART-WRF). The authors demonstrate that the BC emitted from Siberian fires were transported by low-pressure system and reached to the upper troposphere over the Arctic. They separately evaluate effects of large-scale and convective precipitation on wet removal and spatial distribution of BC.

This study is interesting and scientifically important. The subject is of great interest to ACP. However, the interpretations of the transport mechanisms and analysis for wet removal of BC are not satisfactory, which are cause for concern (see Major comments). Another concern is that the manuscript is too long and the authors should attempt to shorten the manuscript (see Specific comments). Most other comments listed below are minor clarifications. Once these points are addressed satisfactory, the paper should in my opinion be suitable for publication in ACP.

**Major comments:**

1.  Section 4.1 and Figure 7a, horizontal poleward eddy heat flux
It seems to me that the author's approach could not represent the activity of the migratory cyclones (WCBs). The authors define the horizontal poleward eddy heat flux ($\overline{v'\theta'}$), where the overbar denotes time-averaging over the ACCESS period (17 days?) and primes are instantaneous deviations from the means (17 days?). In general, a time scale of the migratory cyclones is about a week and it is shorter than the ACCESS period. Application of the long averaging period to the deviations (primes) may not detect the activity of migratory cyclones (WCBs) and anticyclones (i.e., the deviations from the 17-day means would not be adequate for the migratory cyclones). For example, the authors can use the 5-day running means for estimate the instantaneous deviations ($v'$ and $\theta'$) instead of the 17-day means, and then calculate time average (overbar) for the ACCESS period.

2. Section 4.1 and Figure 7, upward BC flux and divergence of horizontal BC flux

The authors state that "a strong ascent of BC mass fluxes are also co-located with areas presenting large values for the divergence of horizontal BC flux in the PBL" (P16L29-30). This would be misleading. According to Oshima et al. (2013), the horizontal convergence represents the upward BC transport from the PBL to the free troposphere, not the horizontal divergence. Considering air flow at the surface, the convergence can uplift air parcels from the surface to the free troposphere (like WCBs), but the opposite divergence cannot. For example, it seems to me that the upward BC region (red color around 90-100E, 50-60N in Fig. 7b) is slightly on the south of the divergence region (Fig. 7c) and corresponds to the light blue convergence region (Fig. 7c), although it is somewhat difficult to read the exact regions from these figures.

The authors estimate upward BC mass flux at the 850-hPa level, but it seems to me that 850-hPa level is too low and 700-hPa is better. For example, Fig. 7b shows that there are large values over north China, but 850-hPa level over this region is close to the ground-level (largely influenced by BC mass concentration, rather than vertical velocity). In addition, because the authors define the PBL as the 700-1000 hPa layer for the horizontal BC flux, it is consistent to use the same 700-hPa level for the vertical BC flux to discuss the horizontal BC transport and the subsequent uplifting from the PBL to the free troposphere.

3. Section 5.3.1, transport efficiency of BC particles ($TE_{BC}$)

I could not understand the advantage of the method (P29L3-6) and why the authors use CO for estimate of $TE_{BC}$ by model. The $TE_{BC}$ values estimated by using modeled CO in Eq. (1) would include uncertainties in CO calculations in the model, as described in section 4.2. The authors state different transport patterns and diffusion during transport (between the BASE and NoWetAll simulations?), but the authors have already used the ratios of two model simulations in Figures 9 and 11 (e.g., NoDry/BASE). I am not sure why the authors do not define $TE_{BC}$ as BC_BASE/BC_NoWetAll, similar as Figures 9 and 11.

The authors define background values of CO in Eq. (1) using CO measurements. I could not understand why the CO background values obtained over the ACCESS flight

regions could be applied to the all model domain. The background values of CO would be different over the Arctic and East Asia. This would cause the uncertainty in estimate of $TE_{BC}$. If the authors use modeled CO in Eq. (1), the use of model background CO values would be better.

In my opinion, the estimation of $TE_{BC}$ using BC/CO ratios would be conducted by observation studies, because they could not estimate BC concentrations not influenced by wet removal (e.g., such as BC_NoWetAll) from the observation. The use of BC/CO ratios for $TE_{BC}$ estimation in Eq. (1) assumes BC-CO correlation over the source regions (similar emission sources for BC and CO). However, the anthropogenic plumes in Figure 8 show some enhancements of BC but little enhancements of CO (no BC-CO correlation?). I am not sure that Eq. (1) could be applied to these plumes, although the CO values would be canceled out in numerator and denominator in Eq. (1).

4.  Sections 5.3.1 and 5.3.2, APT and ACWT calculations and interpretations

It seems to me that the author's APT and ACWT approaches include below-cloud scavenging and in-cloud scavenging (nucleation scavenging with subsequent removal by precipitation, rainout) processes and could not distinguish these two processes, although these approaches could indicate the importance of precipitation on BC removal. It is expected that the both approaches would give the similar results, because the sum of the rain, ice, snow and graupel precipitation rates (used for APT) and the sum of cloud liquid water, ice, snow, rain and graupel contents (used for ACWT) would correlate. The authors should state that the similar results (Fig. 13) obtained from two different approaches suggest the validity of the importance of precipitation on BC removal, rather than effects of nucleation scavenging (P30L3).

**Specific comments:**

P1L13, P13L6, P16L2, P18L13, P18L24, P31L20, P31L21, P33L4, P33L23, Remove "very".

P3L26, "ACCESS campaign", Please spell out it and add a brief description about the campaign here.

P7L1, "size bins (8 in this study)", Please show the minimum and maximum size ranges.

P7L18, Please show the horizontal and time resolutions of the fire inventory.

Section 2.2.2, There is no description about the flaring emissions. Please add a brief description.

P8L19-20, "using the meteorological fields from the WRF-Chem simulation." Is it BASE calculation?

Figures 2 and 3, It is difficult to find the vertical bars (median values). Please make the median values more visible. Please clarify in the figure captions that all flight data are used in these figures.

P10L19-20, "wrong OH and transport", This is not clear what you mean. Do you mean that OH calculation in the CBM-Z scheme has a problem for 6-9 km range but it is OK for other altitude range? I cannot understand this. Please clarify.

P11L19-20, The authors show the overestimation of BC in the mid-troposphere for the Run100 simulation in Figure 3. Please specify the altitude ranges of the overestimation.

P11L21, " This suggests that, at a coarser resolution, the model is unable to resolve the fine structure of plumes transported in altitude", If the authors state this, it is better to check the CO concentrations. Is there overestimation of CO in the mid-troposphere for the Run100 simulation?

P12L5, "Global models always overestimated BC mass", I do not think "always". Some models underestimate BC mass concentrations at upper troposphere during the ARCTAS campaign (Koch et al., 2009).

P13L12, The authors state that the AOD underestimation is due to the simplified SOA calculation. If so, please explain this in more detail.

P16L6-11, "The major objectives of this section ... towards the Arctic.", The authors need not to describe objectives in each section. Please remove this paragraph to shorten

the manuscript.

P16L15, " the overbar denotes time-averaging over the ACCESS period", Please specify the time-averaging period (i.e., the ACCESS period). From 4 July to 21 July in 2012 (17 days)?

P18L11, "Andøya and Spitsbergen", It is better to write the longitude and latitude of these locations.

P18L19-20, "This is mostly due to numerical diffusion in the model", Please explain the numerical diffusion in more detail.

P18L33 and Figure 9, "relative contributions", BC or CO? Please clarify in text and figure captions.

Section 4.4, "four plumes (boreal fires), two plumes (anthropogenic), two plumes (flaring)". It is better to mark (e.g., by circles) these plumes in some of Figure 8. It seems to me that some plumes were observed by the aircraft, but others (e.g., flaring) were not observed (not along the flight tracks). Please clarify these differences in text, because the FLEXPART calculations were conducted for these eight plumes and the reliability of interpretation will be different whether the plumes were observed ones or not (only modeled ones).

Section 4.4, BC and CO concentrations in the anthropogenic and flaring plumes, It seems to me that some enhancement of BC with little enhancement of CO was observed in the anthropogenic plumes. Could you explain this difference? Figures 8 and 9 also show that both BC and CO concentrations were low in the flaring plumes. Could you explain why CO concentrations were low for the anthropogenic and flaring plumes?

P23L2, "(Fig 7)"?

Figure 10d, The authors state that "Heating of large Siberian fires can inject CO and BC into the free troposphere (P21L9)", but Figure 10d shows that high BC concentrations initially appear at 0-2 km in altitude. Please clarify the injection height of the fire emission in text.

P24L8-9 "we use the normalized differences between the NoDry, NoWet, NoWetCu simulations and the BASE run." It is better to express the calculation method to estimate the relative contributions, specifically.

P24L20, "European influence", European anthropogenic influence?

P24L28-29, "An understanding of the wet removal of BC ... to the Arctic.", This means a general importance of wet removal. Please remove this sentence to shorten the manuscript.

P24L31-P26L4 and P33L16-18, The authors state that BC particles were coated with sufficient water-soluble compounds, but they have not shown the coating information for the observed plumes. If the coating information will be available from the SP2 measurements, this information may be helpful for the interpretation, although a portion of thickly-coated BC particles in the observed plumes had been removed by precipitation during transport from the source regions to the Arctic.

P26L22, "map of the upward BC flux and the patterns of the divergence of horizontal BC flux in PBL (Fig.7)." Please see Major comments and remove "the upward BC flux and".

P27L2-3, " BC particles are removed through the nucleation scavenging mechanism or through below-cloud scavenging", Nucleation scavenging of aerosols alone is not a deposition process, because if only nucleation scavenging takes place (aerosols become cloud droplets) and subsequent cloud evaporation takes place, the aerosols would remain there. Or do you mean in-cloud scavenging (rainout)?

P27L18-19, "the below-cloud removal efficiencies are indeed very small." If the authors did not estimate the effects of below-cloud scavenging by the model, please add some references here. Below-cloud scavenging depends on size distributions of particles and precipitation intensity. Is intensity of convective precipitation small?

P28L14, background values of BC mass mixing ratios "and CO concentrations", respectively.

Figure 13, Please mark or emphasis the starting points (release time = 0) of the eight

plumes, if possible.

P30L2-3, "as a function of ACWT, suggesting that BC can also be removed efficiently by nucleation scavenging when transported to the Arctic." Please see Major comments. I could not understand why this result indicates the BC removal by nucleation scavenging. Please explain interpretations of the ACWT results more clearly.

P30L29 and Figure 14, "mean BC mass concentrations zonally averaged during the ACCESS period", The model domain shows that the longitude range used for the zonal averages is different depending on latitude, for example, all longitude range at high latitude but only Asian longitude range at mid latitude. This may contribute to the contrast between the mid-latitude and the Arctic. If so, please add descriptions about the possible effects for the zonal averages due to the different longitude ranges in text.

P31L4, "illustrating the sharp meridional gradient in the distribution of moisture and precipitation", The authors have not shown any results or discussions of moisture. It seems to me that this description will be misleading, because effects of wet removal will be greater where moisture and precipitation are greater (e.g., Asian regions), but $TE_{BC}$ is not smaller over these regions. The $TE_{BC}$ will be smaller for air experiencing wet removal over Asia and that subsequently transported to the outflow regions (high latitude). Please clarify the interpretation.

P31L19, "The interactions between aerosols and clouds", Please clarify what this means.

P33L5-6, "the spatial distribution of the mean upward BC mass fluxes", Please see Major comments and this should be removed.

---

## Author Comment (AC1) · 5 May 2017

The authors would like to thank the Reviewer#1 for his careful review of our manuscript. We addressed each comment individually in the following electronic supplement, and have revised the manuscript accordingly.

*RC : This paper presents a thorough and detailed model investigation of the factors affecting transport of several plumes of BC and CO to the Arctic, comparing the influences of different pollution sources and the spatial distribution, amount and types of precipitation. There is a limited comparison to aircraft measurements that shows the model is doing a reasonable job of capturing the general features of the observations, though not doing a perfect job of simulating the exact concentrations or resolving the finer structure of the plumes (and I would not expect it to). Overall I think the work is of a very high scientific quality, and the main issue I have is that it is too long. However, I think it is probably not suitable to be split into 2 papers, nor are there any sections that could easily be cut without detracting from the rigour of the analysis, so I think it will just have to remain long. Possibly, parts of Section 2 might work as an appendix/supplement. Additionally, some of the written English is phrased in strange ways and some sentences do not quite make sense, so it could do with some work. I recommend publication in ACP subject to the following minor revisions.*

**AC : We thank the anonymous reviewer for providing helpful comments. As mentioned by the reviewer himself/herself, it is not suitable to be split into 2 papers as it would compromise the second paper. We have tried to reduce parts of the paper, as suggested by Reviewer#2, but the length is still important.**

— *RC : P1L13-15 The way this is phrased makes it sound like the source determines the APT. I think this is because you have said "is" rather than "was", which makes it sound more like a general statement rather than a specific statement about the plumes studied in this paper. This type of error occurs throughout the paper and is mostly benign but can sometimes be confusing, such as in the example above. As a general rule of thumb, the work you have done is in the past (e.g. "the campaign took place" or "we ran the model") but things you do in the paper itself should be in the present (e.g. "in this study we describe..." or "Figure 5 shows ..."). It gets more tricky, for example when you come to the conclusions- the specific plumes you studied were affected by precipitation but plumes in general are affected by precipitation.*

**AC : We agree with the reviewer that using present may give the impression of a general comment and may be confusing for the reader. We put the phrase in the past tense. More generally, we have decided to apply the reviewer's suggestion along the full manuscript. We use present when it refers to the work we have performed and the results obtained, and we use the past when statements refer to conclusions.**

— *RC : P3L8-9 "Schwarz et al showed...." I'm not sure what you are trying to say with this sentence, other than to mention the study by Schwarz et al. It seems like you do a better job of saying the same thing in the next sentence, so this one isn't really necessary*

**AC : We removed that sentence.**

— *RC : P3L15 What results from Koch and Hansen? Do they confirm them or do they just agree with them? (i.e. both could be wrong)*

**AC : Koch and Hansen (2005) highlighted an overestimation of BC concentrations in the upper Arctic troposphere. The modeling study of Breider et al. (2014) found similar results. We modify the word "confirms" by "agrees with".**

— *RC : P5L6-10 Please give details of how the SP2 was calibrated. Actually, you should also say how the CO box was calibrated. Just one sentence for each would probably be sufficient if a standard method was used.*

**AC : The SP2 was calibrated using the recommended calibration material (Fullerene soot, as produced, Lot#F12SO11, Alfa Aesar Inc., Ward Hill, MA). CO was measured every second from VUV fluorescence with an accuracy of 10% (Gerbig et al., 1999).**

— *RC : P5L14-15 "Absolute uncertainty of BC particle mass is within 10%, the uncertainty of the derived total BC mass mixing ratio is about 30%." I am not sure I follow the logic here. Also 10% is not an absolute uncertainty, it is a relative uncertainty- a percentage is relative by definition. Given you do not mention particle size in your analysis, the only relevant errors are A) The systematic uncertainty in your BC calibration and B) The statistical uncertainty in the derived BC mass concentrations. A) is down to a combination of the sampling time, concentrations and flowrate, but is easy as you can just pick a time when you think your concentrations are constant and look at the variation in your time series. You could express this as a relative error (e.g. +/- 10%) or absolute (e.g. ± 2 ng/kg) B) is more difficult as you have two factors-firstly the random variation in your calibration slope (in other words how accurately does your particular slope recreate the mass of the calibration material, if you repeated your calibration exactly how much would the slopes differ?) (see Laborde et al. (2012b)) and secondly how well does your calibration material represent the instrument response to the BC you measure in the atmosphere? As Laborde et al. (2012a) showed, the SP2 responds differently to different BC types. I understand the observations present a fairly minor part of your paper but if you are going to present them and quote an uncertainty it should be done correctly, which at present I don't think it is.*

**AC : This was indeed a mistake. The instrument provides acumulation model refractory BC mass mixing ratio with a total relative uncertainty of $30\%$ (Laborde et al., 2012, Schwarz et al., 2013). The recent paper by Schwarz et al. (2017) details how the instrument had been calibrated and how the measurements had been corrected for the ACCESS campaign.**

— *RC : P7L23 You note that the height of the emissions injection is very important- how good a job does the plume rise model do? How does it work out the buoyancy of a particular fire? Please provide a brief summary*

**AC : During the ACCESS airborne campaign, flights were only performed in the remote Arctic region. Validating the fire injection heights would require measurements of vertical profiles of BC over the source regions. The performance of the plume rise model has nevertheless been studied in detail in some papers (Grell et al., 2011 ; Sessions et al., 2011). They have shown that the plume-rise model embedded in WRF-Chem improves the injection heights when compared to the satellite-observed ones. These two papers are quoted in the manuscript. Appropriate fire properties are obtained from a synergy between remote sensing observations, land use and carbon fuel datasets to determine in which columns the fires are located and the plume rise is simulated explicitly (Grell et al., 2011).**

— *RC : Figure 2a is there a reason why potential temperature is more useful than just temperature?*

**AC : The potential temperature is constantly increasing with altitude, suggesting strong atmospheric stability in this study. It has also been used to underline in**

Fig. 10d and Fig. 14 that the transport of BC in biomass buring plumes followed isentropes. Validating this parameter is therefore useful.

— *RC : P10L20 wrong OH and transport- the way this is written it is not quite clear what you mean. Wrong transport ? Vertical or horizontal ? Or just transport ? Do these factors and the studies you reference explain why CO is underestimated specifically between $6 - 9km$ ?*

**AC : This sentence was unclear. It was not referring to WRF-Chem. We have re-written it to be clearer : The small underestimation in CO between 6 and 9 km is a common feature observed by most models (Emmons et al., 2015 ; AMAP, 2015). Variability in models, run with the same emissions, appears to be driven by differences in chemical schemes influencing modelled OH and/or differences in modelled vertical export efficiency of CO from mid-latitude source regions to the Arctic (Monks et al., 2015).**

— *RC : Section 3.2 I think you are overselling the agreement between model and measurements. For example "The two profiles are well correlated with maximum CO values of 200 ppbv at $7 - 8$ km, associated with elevated BC values reaching 25 ng $kg^{-1}$." Actually the maximum in the model CO is ~150 ppb at 6.5km. Compare to P32L10 : this is a better way of describing the model/measurement comparison. The general features were well captured.*

**AC : The rewiewer might have been confused about that sentence. We just compare here the two measured profiles of CO vs BC, not model vs observations. This comparison is done later, e.g. P11L8. We have added the word "the two measured profiles" to avoid confusion.**

— *RC : P11 L 1 "the influence of flaring emissions in this area is insignificant". Insignificant for what ? At what scale ? Later on in the paper you talk about some flaring plumes so it can't have been insignificant within those plumes.*

**AC : The term "in this area" was not clear. The influence of flaring emissions on the vertical profiles of CO and BC sampled by the aircraft is insignificant. It has however a small influence on the background pollution off the coast of Scandanavia, as simulated by the model but not in the plumes sampled by the aircraft. According to a comment of Reviewer#2, we have decided to removed this sentence.**

— *RC : P11L7-8 "The model shows appreciable skill in capturing the vertical profile of BC, but overestimates the BC mixing ratio between 2 and 3 km of altitude." Looking at the median BC concentrations, the 40km model overestimates between $1.5 - 7.5km$ then underestimates higher than 7.5km. Additionally, if you just went with a flat BC concentration of around 6 or 7 ng/kg the medians would probably show similar agreement. Now, I am not saying that the agreement is terrible- actually it OK, perhaps reasonably good, and it seems to capture the general features of the observations without doing a perfect job. But you read the text and it sounds like the model is doing an amazingly good job, which figures 3 and 8 show it isn't, it's just doing a reasonable job. So I think you should just tone down how well you claim the model and observations agree.*

**AC : We modified the sentence as follows : "The model shows appreciable skill in capturing the general structure of the vertical profile of BC, but overestimates the BC mixing ratio in the mid-troposphere".**

— *RC : P11L9 The 30% error in the SP2- please relate this back to the previous comment on statistical vs systematic error. Also please give numbers for the model biases.*

**AC : The resulting mean bias on BC is 1.5 ng kg$^{-1}$ and the corresponding normalized mean bias if 27%. It is much lower that biases reported for most models in the Arctic region (AMAP, 2015). Eckhardt et al. (2015) indeed reported that BC concentrations in July-September are overestimated in the mean of intercompared models by 88%.**

— *RC : P11L13 Here you say the CO between 6-9km is due to biomass burning emissions, but previously you said the underestimation in CO was due to wrong OH concentrations and transport. Could it not be that the model is underestimating the biomass burning CO at this altitude? Is that what you mean by transport? Please clarify*

**AC : The underestimation in CO mentioned previously was a general features in CTM. Here we wanted to mention the origin of the plumes we sampled in this ACCESS study. For sake of clarity we modified the last sentence : "In Sect. 4, we discuss the origin and transport of plumes leading to this noticeable increases of CO and BC between 6 and 9 km of altitude, associated with higher ozone mixing ratios."**

— *RC : P13L11 You say the AOD underestimation is due to simplified SOA, but you haven't given any details of how the AOD is calculated. I don't think you can say this is due to missing SOA when you don't have a good handle on even the size distribution, let alone composition or refractive index.*

**AC : This is correct. We don't know the exact reason of the model underestimation in this region. AOD is computed through a Mie code embedded in the model. The representation of the size distribution and complex refractive index strongly influences the result. The simplified SOA mechanism is a potential cause, but we can't say it is the main one. We removed that comment.**

— *RC : P18L13 are the BC enhancements you are talking about in the model or in the observations? I assume model but it's not clear*

**AC : We added the word "modeled".**

— *RC : P18L23 The agreement between model and measured BC is "very good"- Again I would say it's reasonable but I wouldn't say it's "very good". The model plumes in figure 5 are too diffuse and some are missing, such as the smaller amounts of BC associated with the CO plume at 8km at 0915. You get the general features. For me, "very good" would be if you could plot model vs measurement for each grid box as the aircraft passed through it and get something approaching a 1 :1 line, though I doubt that could happen in a study like this.*

**AC : We corrected the sentence, replacing "very good" by "reasonable".**

— *RC : Figure 8 You could do with a longer time average of the measurement data as currently it's difficult to see the structure when the markers all overlap. I also wonder if there is a way to make the aircraft easier to see as the parts that stand out are the parts where it disagrees with the model. Also it might help if the x-axis was north/south or east/west as several points in the discussion relate to spatial location and this is difficult to see in this and subsequent plots*

**AC : We follow the reviewers's suggestion in applying a sliding window to calculate longer time average of the measurement data. In situ measurements are averaged**

using a 2-min running mean. We have added larger white dots behind the measurements to make them easier to see versus the modeled cross-sections. We have also highlighted by magenta circles the eight airmasses discussed in the Sect. 4.4 to facilitate the reading.

— *RC : P21L18 If the aerosol from flaring had been removed by precipitation, wouldn't you still see the CO ?*

**AC : We thank the reviewer for pointing this. Indeed, if the decrease of BC in flaring plumes was only due to precipitation, the CO should have remained unchanged. This is not the case, suggesting that transport of plumes from flaring sources is not only directed towards the northern coast of Norway. We have decided to remove this last sentence.**

— *RC : P21L25 Didn't you say in the previous section that the flaring plumes didn't exist ?*

**AC : No, we had said previously that the aircraft did not sample any flaring plume. But flares have a small influence in the area of the study (between northern Norway and Svalbard archipelago). Those flaring airmasses have been identified in Fig. 8 and 9 and highlighted by circles in the new version of the manuscript.**

— *RC : Section 4.4 The discussion may be easier to follow here if you circled the plumes on one of the figures*

**AC : We agree. This has been done in Fig. 8.**

— *RC : P23L2 you say the fire injection lofted the plumes to 6km, but doesn't fig 10d shows that the emissions from the 8th initially remained below 4km. I'm not really sure I follow what figure 10d adds to the analysis*

**AC : This is correct, thanks for pointing this. The rapid uplift up to 6 km is due to WCBs not pyroconvection. The sentence has been removed. Fig. 10d illustrates the merge of the two BB over eastrn Siberia on 12 july and their transport to the Arctic following isentropes.**

— *RC : P24L20 How does figure 9 show European influence ?*

**AC : Below** $4$ **km, the BC concentrations are dominated by the anthropogenic contribution (Fig. 9). As discussed in Sect. 4.4, this area is influenced by European emissions. To clarify the sentence, we have written "European anthropogenic influence".**

— *RC : P24L24 Please define KFCuP in the text. You refer several times to KFCuP as if it is a process itself. In the real atmosphere it is actually "convective clouds" that do things that the model is trying to represent.*

**AC : We had defined KFCuP in Table 1. We replaced "KFCuP" occurences by "convective clouds" when the manuscript speaks about processes and keep "KFCup" only when the cumulus scheme is described.**

— *RC : P26L4 It's frowned upon these days to refer to it as a coating, as that may not represent the actual morphology of the particle. Maybe say BC in BB plumes is more internally mixed or*

*something like that. I also saw another point where you referred to coatings, please also change this.*

**AC : This has been corrected P24L31, P28L3 and P33L16.**

— *RC : P26L11-14 I don't know what these last 2 sentences add to the analysis*

**AC : We have noted that the wet deposition processes were efficient in removing BC-containing particles during transport. But in Fig. 6, we had shown that clouds in this study are mostly mixed-phased or ice clouds. The last two sentences have been kept to quote studies underlying the role of impaction scavenging in removing BC in mixed-phased or ice clouds.**

— *RC : P27L19 The removal efficiencies may be low for large rain drops but not for drizzle*

**AC : Here we are talking about cumulus parameterized clouds. Drizzle mostly occurs in low-level stratocumulus clouds that are resolved by the model and do not depend on KFCuP mechanism.**

— *RC : P29L6 The 5th percentile of the measured or the modelled CO concentration? Also why not do this for the BC as well? The average lifetime of aerosols is of the order of a week because of deposition processes, mostly wet deposition. So the lifetime of aerosol that escapes wet deposition is longer. If you mean you looked at figure 3 and saw that the minimum BC in the observations was basically zero at all levels then say that. But if that was the case I still don't see the harm in taking the 5th percentile like you do with the CO.*

**AC : This is correct. But according to a major comment of Reviewer#2, we have modified the calculation of the transport efficiency of BC using the ratio of BC in the base run on BC in the NoWetAll run. This is more appropriate for a model estimate. Therefore, the background concentrations are not used anylonger.**

— *RC : P29L22 Perhaps this is not the case in the model but in the real atmosphere BC would only be lost if the cloud precipitated. If it evaporated the BC would still be there.*

**AC : We agree with the reviewer. However we do not consider losses for the *atmosphere* but rather losses for the *plumes*. When there is activation of clouds droplets, BC is not lost for the atmosphere but is transferred from interstitial aerosol to cloud-borne aerosol (nucleation scavenging). If the cloud droplets reach the sizes of precipitating rain drops, it will act as a deposition process from the plume.**

— *RC : P32L16 Can you suggest how the discrepancy/difference might be resolved? Is it just because there is less precipitation in this study?*

**AC : We have suggested three reasons to explain the differences between the two papers : different year (BC emissions differ), less precipitation and a difference in the methodology. Matsui's study is only based on observations. The BC-to-CO ratios are therefore normalized by BC-to-CO ratios over the sources. If a plume emitted from a specific source diverges before reaching the receptor area, the transport efficiency calculated by Matsui's method will decrease. In our modelling study, the transport efficiencies are only influenced by precipitation and therefore higher. They illustrate the role of deposition during transport rather than the contribution of the transport itself.**

— *RC : P33L30 "found to be more important" It is not clear what you mean by this. If you mean the cumulus clouds remove more of the low-level aerosol by scavenging or washout than by uplifting it then say that. It's better to say what is actually happening.*

**AC : Yes, the reviewer exactly understood the message. We modified the sentence accordingly.**

— *RC : P34L1 This last sentence is an odd way to end a paper. It sounds like you are using the model to validate the measurements when actually it was the other way round. Also this wasn't the main focus of your analysis. It seems to me that the main important conclusions were 1) BC is transported more efficiently into arctic from high-latitude BB sources than from east-Asian anthropogenic sources because it rains less at higher latitudes 2) The ways in which the large-scale vs subgrid convective clouds affected the BC distributions differently. It would be good if you could highlight these more in the abstract/conclusions in terms of the physical processes your results suggest are actually taking place in the real atmosphere, rather than abstract terms like grid-scale and APT*

**AC : The last sentence of the paper has been removed. We have reformulated the abstract and conclusions in focusing on the physical processes : we have for instance replaced "grid scale precipitation" by "large scale coulds" and "subgrid parameterized clouds" by "convective updrafts".**

— *RC : Finally, given you have actual observations it seems like there is a missed opportunity to calculate TEBC based on the measured values of BC and CO. You have hinted at your reasons for not doing so in the comparison to previous studies calculating TEBC but I'm not sure it's clear exactly why not.*

**AC : We have only observations over northern Norway. Calculations of TEBC based on measurements would require also measurements of CO and BC over the different emission sources (anthropogenic, fires and flaring).**

Technical corrections

— *P3L25 "lead to in the" remove 'in'*

**AC : Done.**

— *P3L27-31 This is a very long sentence, please split it into at least 2.*

**AC : Done.**

— *P8L4 add "resolution" before "to adequately"*

**AC : Done.**

— *P11L12 remove "but"*

**AC : Done.**

— *Figure 3 please make the plots slightly taller to show the detail better*

**AC : The vertical bars for the median values have been replaced by diamonds of**

the same color as the mean values. This makes the plot clearer.

— *P11L19 This is a very long sentence, please split it into at least 2.*

**AC : P11L19 is very short. Maybe the reviewer wanted to point P11L13-16. This long sentence has been removed according to a previous comment.**

— *P12 L4 remove "that"*

**AC : Done.**

— *Figure 4 the scale is between 0 – 1 but the text mentions values up to 2.5*

**AC : This is true. But we have chosen this color scale on purpose to hightlight the contrast between the source regions and the rest. The colorbar contains arrows on the top and bottom of it to show that data exist above (resp. below) the highest (resp. lowest) contour level.**

— *Figure 5 are (a) and (b) really the same grid? Panel (a) looks much more blocky. Also label (%) on panel (c) colorscale*

**AC : Fig. 5a is provided by GPCP data at $1°$ resolution. They have been interpolated on the WRF grid, which gives this blocky feature. We have removed "same grid" to avoid confusion. We have also added the missing label.**

— *P15L1 "than in altitude", do you mean "that at higher altitudes"?*

**AC : Done.**

— *P18L6 "plumes transported in altitude" do you mean "at altitude"?*

**AC : We wanted to say "in the upper troposphere". This has been corrected.**

— *P18L20 "shifted a bit"- how far? Saying "a bit" is probably a bit too colloquial for a paper (but not a review!)*

**AC : To be more quantitative, the plume enriched in CO is shifted towards the north by one pixel in the model simulation, which means $40$ km. It has been precised in the new version.**

— *P21L17 "not due to" do you mean "did not lead to"?*

**AC : Done.**

— *P27L34-35 This sentence doesn't make sense*

**AC : The new sentence is the following : "The combination of low-level scavenging in the Arctic region and transport decrease from mid-latitudes is the cause of the low summertime BC concentrations."**

— *Figure 13 caption Maybe rephrase to "The points are colored by the time (in hours) before the release of the trajectories in FLEXPART-WRF)"*

**AC : Done.**

— *P30L25 You may consider not using the word "septenrional" as it is not a commonly used word.*

**AC : We say that BB sources are located far north.**

— *P32L4 Again change "a bit" to "slightly"*

**AC : Done.**

— *P33L4 Remove "very"*

**AC : Done.**

— *Finally, thankyou for your interesting (but long!) paper*

**AC : We also thank the anonymous reviewer for providing helpful comments.**

---

## Author Comment (AC2) · 5 May 2017

The authors would like to thank the Reviewer#2 for his careful review of our manuscript. We addressed each comment individually in the following electronic supplement, and have revised the manuscript accordingly.

General comments :

*RC : This work investigates the transport processes of black carbon (BC) from Siberia to the Arctic and the effects of wet removal over mid- and high-latitudes during the ACCESS aircraft campaign using a regional scale chemical transport model (WRF-Chem) and a Lagrangian particle dispersion model (FLEXPART-WRF). The authors demonstrate that the BC emitted from Siberian fires were transported by low-pressure system and reached to the upper troposphere over the Arctic. They separately evaluate effects of large-scale and convective precipitation on wet removal and spatial distribution of BC. This study is interesting and scientifically important. The subject is of great interest to ACP. However, the interpretations of the transport mechanisms and analysis for wet removal of BC are not satisfactory, which are cause for concern (see Major comments). Another concern is that the manuscript is too long and the authors should attempt to shorten the manuscript (see Specific comments). Most other comments listed below are minor clarifications. Once these points are addressed satisfactory, the paper should in my opinion be suitable for publication in ACP.*

**AC : We thank the anonymous reviewer for his/her careful reading of the paper and for providing helpful comments, which help to improve the quality of the manuscript.**

Major comments :

1. *RC : Section 4.1 and Figure 7a, horizontal poleward eddy heat flux It seems to me that the author's approach could not represent the activity of the migratory cyclones (WCBs). The authors define the horizontal poleward eddy heat flux ($\overline{v'\theta'}$), where the overbar denotes time-averaging over the ACCESS period (17 days ?) and primes are instantaneous deviations from the means (17 days ?). In general, a time scale of the migratory cyclones is about a week and it is shorter than the ACCESS period. Application of the long averaging period to the deviations (primes) may not detect the activity of migratory cyclones (WCBs) and anticyclones (i.e., the deviations from the 17-day means would not be adequate for the migratory cyclones). For example, the authors can use the 5-day running means for estimate the instantaneous deviations ($v'$ and $\theta'$) instead of the 17-day means, and then calculate time average (overbar) for the ACCESS period.*

   **AC : In the original version of the manuscript, the time-averaging was indeed performed over the full ACCESS period (17 days) and the primes were instantaneous deviations from the 17-days means. The goal was to illustrate the persistence of the cyclones over northern Russia and Siberia where they are formed before reaching the Arctic region. We agree with the reviewer that the proposed approach is a better way to highlight the activity of migratory cyclones due to their shorter lifetimes. We therefore follow this advice and use the 5-day running means to assess the instantaneous deviations and then calculate time average for the ACCESS period (17 days). The result is shown on updated Fig. 7a. The main outflow regions remain exactly the same. The advantage of this method is that it underlines that values of the poleward eddy flux were larger over the Sakha (Yakutia) Republic region than in other outflow areas, illustrating the crucial role of this region in exporting fire plumes during summer.**

2. *RC : Section 4.1 and Figure 7, upward BC flux and divergence of horizontal BC flux The authors state that "a strong ascent of BC mass fluxes are also co-located with areas presenting large values for the divergence of horizontal BC flux in the PBL" (P16L29-30). This would be misleading.*

*According to Oshima et al. (2013), the horizontal convergence represents the upward BC transport from the PBL to the free troposphere, not the horizontal divergence. Considering air flow at the surface, the convergence can uplift air parcels from the surface to the free troposphere (like WCBs), but the opposite divergence cannot. For example, it seems to me that the upward BC region (red color around 90-100E, 50-60N in Fig. 7b) is slightly on the south of the divergence region (Fig. 7c) and corresponds to the light blue convergence region (Fig. 7c), although it is somewhat difficult to read the exact regions from these figures. The authors estimate upward BC mass flux at the 850-hPa level, but it seems to me that 850-hPa level is too low and 700-hPa is better. For example, Fig. 7b shows that there are large values over north China, but 850-hPa level over this region is close to the ground-level (largely influenced by BC mass concentration, rather than vertical velocity). In addition, because the authors define the PBL as the 700-1000 hPa layer for the horizontal BC flux, it is consistent to use the same 700-hPa level for the vertical BC flux to discuss the horizontal BC transport and the subsequent uplifting from the PBL to the free troposphere.*

**AC : The text was really misleading. We had written "PBL" instead of "FT", which was very confusing for the reader. We apologize for that. The strong ascent of BC mass fluxes are indeed co-located with areas presenting large values for the divergence of horizontal BC flux in the FT (Fig. 7d) and with areas presenting large values for the convergence of horizontal BC flux in the PBL (Fig. 7c). Convergence of BC flux in the lowest layers can uplift air parcels from the surface to the free troposphere, which are then exported to ouflow regions (divergence of BC flux in the FT). We also agree with the reviewer that it is more convenient to plot the upward BC flux at 700 hPa (top of the so-called PBL) rather than at 850 hPa to avoid the contamination by the emissions. It has been done in the new version. We have also changed the color scale of Fig. 7c and 7d to better highlight the convergence areas in the PBL (blue). The result is much clearer with a very good spatial correlation between the regions with large upward BC flux, convergence (resp. divergence) of the BC horizontal flux in the PBL (resp. FT). The results remain unchanged.**

3. *RC : Section 5.3.1, transport efficiency of BC particles (TEBC) I could not understand the advantage of the method (P29L3-6) and why the authors use CO for estimate of TEBC by model. The TEBC values estimated by using modeled CO in Eq. (1) would include uncertainties in CO calculations in the model, as described in section 4.2. The authors state different transport patterns and diffusion during transport (between the BASE and NoWetAll simulations?), but the authors have already used the ratios of two model simulations in Figures 9 and 11 (e.g., NoDry/BASE). I am not sure why the authors do not define TEBC as $BC_{BASE}/BC_{NoWetAll}$, similar as Figures 9 and 11. The authors define background values of CO in Eq. (1) using CO measurements. I could not understand why the CO background values obtained over the ACCESS flight regions could be applied to the all model domain. The background values of CO would be different over the Arctic and East Asia. This would cause the uncertainty in estimate of TEBC. If the authors use modeled CO in Eq. (1), the use of model background CO values would be better. In my opinion, the estimation of TEBC using BC/CO ratios would be conducted by observation studies, because they could not estimate BC concentrations not influenced by wet removal (e.g., such as $BC_{NoWetAll}$) from the observation. The use of BC/CO ratios for TEBC estimation in Eq. (1) assumes BC-CO correlation over the source regions (similar emission sources for BC and CO). However, the anthropogenic plumes in Figure 8 show some enhancements of BC but little enhancements of CO (no BC-CO correlation?). I am not sure that Eq. (1) could be applied to these plumes, although the CO values would be canceled out in numerator and denominator in Eq.(1).*

**AC : As noted by the reviewer, the estimation of TEBC using BC/CO ratios has also**

been conducted by a few studies in other regions. Using this proxy that has already been used was clearly a motivation to propose a comparison of values obtained in this study (we compared our results to Matsui's values for example). Notwithstanding, we understand that this proxy has been defined for observation studies, not modelling ones. We have therefore preferred in the new manuscript the definition of TEBC defined by Oshima et al. (2013), which enables to get rid of errors linked to CO uncertainties. Those errors were however limited by the fact that CO values in Eq. (1) were almost canceled out in numerator and denominator. The background CO is not necessary anylonger. Figures 13 and 14 have been modified accordingly. The results are identical to our previous results, except when APT values are very small : TEBC were not equal to $100\%$, but close to $95\%$, in the original manuscript due to errors ascribed to CO uncertainties. This artefact has vanished in the updated paper.

4. *RC : Sections 5.3.1 and 5.3.2, APT and ACWT calculations and interpretations It seems to me that the author's APT and ACWT approaches include below-cloud scavenging and in-cloud scavenging (nucleation scavenging with subsequent removal by precipitation, rainout) processes and could not distinguish these two processes, although these approaches could indicate the importance of precipitation on BC removal. It is expected that the both approaches would give the similar results, because the sum of the rain, ice, snow and graupel precipitation rates (used for APT) and the sum of cloud liquid water, ice, snow, rain and graupel contents (used for ACWT) would correlate. The authors should state that the similar results (Fig. 13) obtained from two different approaches suggest the validity of the importance of precipitation on BC removal, rather than effects of nucleation scavenging (P30L3).*

**AC : Both APT and ACWT approaches include below-cloud and in-cloud scavenging. The difference between the two approaches refers to the nucleation scavenging process. To compute APT and ACWT, precipitation rates for the different hydrometeors are summed *only* in WRF-Chem grid boxes crossed by the FLEXPART-WRF trajectories. If activation occurs, particles are transferred from interstitial aerosol phase to cloud-borne aerosol phase (nucleation scavenging). As far as APT is concerned, this is not a deposition process if cloud droplets do not grow enough to produce rain drops that precipitate. But it can be considered as a deposition process for the plume if the droplets containing aerosols reach the sizes of precipitating rain drops in other cells (for example above or below the considered grid box crossed by the trajectory). In that case, the sum of cloud liquid water and ice crystals mixing ratios in such grid boxes is relevant. This is the purpose of ACWT. The sum of the rain, ice, snow and graupel precipitation rates (used for APT) and the sum of cloud liquid water, ice, snow, rain and graupel contents (used for ACWT) do not correlate. This correlation strongly depends on the size distribution of hydrometeors. Nevertheless, we agree that the two approaches almost give similar results, except for the flaring plumes, as indicated in the paper.**

Specific comments :
— *RC : P1L13, P13L6, P16L2, P18L13, P18L24, P31L20, P31L21, P33L4, P33L23, Remove "very".*

 **AC : Done.**

— *RC : P3L26, "ACCESS campaign", Please spell out it and add a brief description about the campaign here.*

**AC : ACCESS stands for Arctic Climate Change, Economy and Society. This has been included in the new version in addition to the location and time of the airborne campaign.**

— *RC : P7L1, "size bins (8 in this study)", Please show the minimum and maximum size ranges.*

**In MOSAIC, aerosol size distribution is between 39 nm and $10 \mu$m.**

— *RC : P7L18, Please show the horizontal and time resolutions of the fire inventory. Section 2.2.2, There is no description about the flaring emissions. Please add a brief description.*

**AC : FINN provides daily, 1 km resolution emissions. The description of flaring emissions was indeed missing : flaring emissions are from the ECLIPSE (Evaluating the CLimate and Air Quality ImPacts of Short-livEd Pollutants) inventory (Stohl et al., 2015).**

— *RC : P8L19-20, "using the meteorological fields from the WRF-Chem simulation." Is it BASE calculation ?*

**AC : yes, it is the BASE run. It has been specified.**

— *RC : Figures 2 and 3, It is difficult to find the vertical bars (median values). Please make the median values more visible. Please clarify in the figure captions that all flight data are used in these figures.*

**AC : To represent the median values, the vertical bars have been replaced by dimaonds of the same color as the mean values. The captions of Fig. 2 and Fig. 3 mention that data have been interpolated along all the 14 Falcon flight tracks.**

— *RC : P10L19-20, "wrong OH and transport", This is not clear what you mean. Do you mean that OH calculation in the CBM-Z scheme has a problem for 6-9 km range but it is OK for other altitude range ? I cannot understand this. Please clarify.*

**AC : This sentence does not refer to the slight underestimation of the CO in this study by WRF-Chem model via CBM-Z scheme. It was a general sentence explaining the general underestimation of CTM in the Arctic. We have re-written it to be clearer : The small underestimation in CO between 6 and 9 km is a common feature observed by most models (Emmons et al., 2015 ; AMAP, 2015). Variability in models, run with the same emissions, appears to be driven by differences in chemical schemes influencing modelled OH and/or differences in modelled vertical export efficiency of CO from mid-latitude source regions to the Arctic (Monks et al., 2015).**

— *RC : P11L19-20, The authors show the overestimation of BC in the mid-troposphere for the Run100 simulation in Figure 3. Please specify the altitude ranges of the overestimation.*

**AC : The overestimation is between 1.5 and 5 km.**

— *RC : P11L21, " This suggests that, at a coarser resolution, the model is unable to resolve the fine structure of plumes transported in altitude", If the authors state this, it is better to check the CO concentrations. Is there overestimation of CO in the mid-troposphere for the Run100*

*simulation ?*

**AC : Yes, we have noted a corresponding overestimation of CO in the mid-trosposphere in the Run100 simulation. There is an enhancement of** 4 **to** 5 **ppbv as compared to the BASE run.**

— *RC : P12L5, "Global models always overestimated BC mass", I do not think "always". Some models underestimate BC mass concentrations at upper troposphere during the ARCTAS campaign (Koch et al., 2009).*

**AC : We replaced "always" by "generally".**

— *RC : P13L12, The authors state that the AOD underestimation is due to the simplified SOA calculation. If so, please explain this in more detail.*

**AC : We don't know the exact reason of the model underestimation in this region. AOD is computed through a Mie code embedded in the model. The representation of the size distribution and complex refractive index also strongly influences the result. The simplified SOA mechanism is a potential cause, but we can't say it is the main one. According to a comment of Reviewer#1, this sentence has been removed.**

— *RC : P16L6-11, "The major objectives of this section … towards the Arctic.", The authors need not to describe objectives in each section. Please remove this paragraph to shorten the manuscript.*

**AC : It has been removed, which helps to shorten a bit the paper.**

— *RC : P16L15, " the overbar denotes time-averaging over the ACCESS period", Please specify the time-averaging period (i.e., the ACCESS period). From 4 July to 21 July in 2012 (17 days) ?*

**AC : The information about the time-averaging has been added.**

— *RC : P18L11, "Andøya and Spitsbergen", It is better to write the longitude and latitude of these locations.*

**AC : We have specified the coordinates of Andoya (**69.1°N, 15.7°E**) and Spitsbergen (**78.9°N, 18.0°E**).**

— *RC : P18L19-20, "This is mostly due to numerical diffusion in the model", Please explain the numerical diffusion in more detail.*

**AC : Numerical diffusion is caused by the finite difference method applied to the advection equation on the model grid. In the model, the gradients are simply taken along coordinate surfaces, hence are imperfectly described. A 6th-order numerical diffusion is used in our simulations.**

— *RC : P18L33 and Figure 9, "relative contributions", BC or CO ? Please clarify in text and figure captions.*

**AC : This had been forgotten. Fig. 9 shows the relative contributions of BC concentrations due to the different emission sources.**

— *RC : Section 4.4, "four plumes (boreal fires), two plumes (anthropogenic), two plumes (flaring)". It is better to mark (e.g., by circles) these plumes in some of Figure 8. It seems to me that some plumes were observed by the aircraft, but others (e.g., flaring) were not observed (not along the flight tracks). Please clarify these differences in text, because the FLEXPART calculations were conducted for these eight plumes and the reliability of interpretation will be different whether the plumes were observed ones or not (only modeled ones).*

**AC : The eight airmasses selected for further analysis have been marked by magenta circles on Fig. 8. We have given them a name for further discussion, distinguishing biomass burning plumes (BB1, BB2, BB3, BB4), anthropogenic airmasses (An1, An2) and airmasses from flaring sources (Fl1, Fl2). We also clarified that only the fire plumes were observed by the aircraft. The anthropogenic and flaring ones have been detected on the model cross-sections.**

— *RC : Section 4.4, BC and CO concentrations in the anthropogenic and flaring plumes, It seems to me that some enhancement of BC with little enhancement of CO was observed in the anthropogenic plumes. Could you explain this difference ? Figures 8 and 9 also show that both BC and CO concentrations were low in the flaring plumes. Could you explain why CO concentrations were low for the anthropogenic and flaring plumes ?*

**AC : The color scales chosen in Fig. 8 might have given the impression that a small enhancement of BF was correlated with little enhancement of CO in flaring and anthropogenic plumes. This is actually not the case. The enhancements in BC and CO are on the same amount of magnitude (in %). If the decrease of BC in anthropogenic and flaring plumes was only due to precipitation, the CO should have remained unchanged. This is not the case, suggesting that transport of plumes from anthropogenic and flaring sources is not only directed towards the northern coast of Norway.**

— *RC : P23L2, "(Fig 7)" ?*

**AC : This was a misprint. Thanks for noticing it.**

— *RC : Figure 10d, The authors state that "Heating of large Siberian fires can inject CO and BC into the free troposphere (P21L9)", but Figure 10d shows that high BC concentrations initially appear at 0-2 km in altitude. Please clarify the injection height of the fire emission in text.*

**AC : This was a mistake and has also been detected by Reviewer#1. The sentence has been removed. The rapid uplift to 6 km is due to WCBs over eastern Siberia.**

— *RC : P24L8-9 "we use the normalized differences between the NoDry, NoWet, NoWetCu simulations and the BASE run." It is better to express the calculation method to estimate the relative contributions, specifically.*

**AC : We have clarified this by giving the formula used for those calculations : $100 \times \left( \dfrac{NoX - BASE}{NoX} \right)$, where NoX represents the NoDry, NoWet or NoWetCu simulation.**

— *RC : P24L20, "European influence", European anthropogenic influence ?*

**AC : Corrected.**

— *RC : P24L28-29, "An understanding of the wet removal of BC ... to the Arctic.", This means a general importance of wet removal. Please remove this sentence to shorten the manuscript.*

**AC : This has been removed.**

— *RC : P24L31-P26L4 and P33L16-18, The authors state that BC particles were coated with sufficient water-soluble compounds, but they have not shown the coating information for the observed plumes. If the coating information will be available from the SP2 measurements, this information may be helpful for the interpretation, although a portion of thickly-coated BC particles in the observed plumes had been removed by precipitation during transport from the source regions to the Arctic.*

**AC : This information would have been indeed very useful. Unfortunately, te coating thickness was not available from the SP2 measurements.**

— *RC : P26L22, "map of the upward BC flux and the patterns of the divergence of horizontal BC flux in PBL (Fig.7)." Please see Major comments and remove "the upward BC flux and".*

**AC : Figure 7 has been updated according to the answer to major comments. The term "PBL" has been replaced by "FT".**

— *RC : P27L2-3, " BC particles are removed through the nucleation scavenging mechanism or through below-cloud scavenging", Nucleation scavenging of aerosols alone is not a deposition process, because if only nucleation scavenging takes place (aerosols become cloud droplets) and subsequent cloud evaporation takes place, the aerosols would remain there. Or do you mean in-cloud scavenging (rainout)?*

**AC : We never say that the nucleation scavenging mechanism was a deposition process, which would suppose that aerosols are removed from the atmosphere by this process. However we do not consider losses for the *atmosphere* here but rather losses for the *plumes*. When there is activation of clouds droplets, BC is not lost for the atmosphere but is transferred from interstitial aerosol to cloud-borne aerosol (nucleation scavenging). If the cloud droplets reach the sizes of precipitating rain drops, it will act as a deposition process from the plume.**

— *RC : P27L18-19, "the below-cloud removal efficiencies are indeed very small." If the authors did not estimate the effects of below-cloud scavenging by the model, please add some references here. Below-cloud scavenging depends on size distributions of particles and precipitation intensity. Is intensity of convective precipitation small?*

**AC : These lines refer to below-cloud scavenging in parameterized cumulus clouds only. The model takes into account below-cloud scavenging in grid-resolved clouds. The intensity of convective precipiatation can be seen in Fig. 5 combining the total precipitation (5b) and the fraction of convective precipitation (5c). The efficiency of below-cloud scavenging depends on the ratio of the sizes of particles and rain drops. This ratio is very small here as mostly all BC-containing particles ar in the fine and accumulation modes $(80 - 470$ nm).**

— *RC : P28L14, background values of BC mass mixing ratios "and CO concentrations", respectively.*

**AC : This has been added. Thank you for pointing this.**

— *RC : Figure 13, Please mark or emphasis the starting points (release time = 0) of the eight plumes, if possible.*

**AC : We have added magenta circles in Fig. 13. They emphasis the starting points of the eright airmasses discussed in Sect. 4.4.**

— *RC : P30L2-3, "as a function of ACWT, suggesting that BC can also be removed efficiently by nucleation scavenging when transported to the Arctic." Please see Major comments. I could not understand why this result indicates the BC removal by nucleation scavenging. Please explain interpretations of the ACWT results more clearly.*

**AC : Please see our response to major comments. When there is some activation of BC-containing particles during the transport of a plume, there is a transfer of interstitial particles to cloud droplets. This suggests a loss of aerosol in the advected plume (but not from the atmosphere if subsequent cloud evaporation takes place). The difference between ACWT and APT is the sum of the integrated mixing ratios of cloud liquid water and ice crystals in clouds. The fact that the relation between $TE_{BC}$ and $ACWT$ is slightly different that the one between $TE_{BC}$ and $APT$, especially in flaring plumes, indicates a role of cloud liquid water and ice crystals in clouds to remove BC during transport. This is caused by the nucleation scavenging mechanism. This has been explained more clearly in the text.**

— *RC : P30L29 and Figure 14, "mean BC mass concentrations zonally averaged during the ACCESS period", The model domain shows that the longitude range used for the zonal averages is different depending on latitude, for example, all longitude range at high latitude but only Asian longitude range at mid latitude. This may contribute to the contrast between the mid-latitude and the Arctic. If so, please add descriptions about the possible effects for the zonal averages due to the different longitude ranges in text.*

**AC : The reviewer is right but it has been done on purpose. Figure 9 had shown that the sum the relative contributions of anthropogenic, flaring and fire emissions to the total BC in the Arctic was higher than $98\%$. This confirms that the influence of the model boundary conditions on Arctic BC at this period is insignificant. The goal of Fig. 14 is to show the relation between BC mixing ratio and $TE_{BC}$ at different latitudes for plumes that have been transported to the Arctic during the ACCESS period. Our study shows that our domain is appropriate to include all sources influencing the Arctic region in July 2012. Some details have been added to the text.**

— *RC : P31L4, "illustrating the sharp meridional gradient in the distribution of moisture and precipitation", The authors have not shown any results or discussions of moisture. It seems to me that this description will be misleading, because effects of wet removal will be greater where moisture and precipitation are greater (e.g., Asian regions), but TEBC is not smaller over these regions. The TEBC will be smaller for air experiencing wet removal over Asia and that subsequently transported to the outflow regions (high latitude). Please clarify the interpretation.*

**AC : We agree with the reviewer, this sentence was confusing. It has been replaced by "This sharp meridional gradient is due to the fact that $TE_{BC}$ is indeed smaller for air experiencing wet removal over Asia or Siberia and that is subsequently transported to the outflow regions (high latitudes)."**

*RC : P31L19, "The interactions between aerosols and clouds", Please clarify what this means.*

**AC : "The interactions between aerosols and clouds" has been replaced by "Aerosol removal in cumulus clouds".**

*RC : P33L5-6, "the spatial distribution of the mean upward BC mass fluxes", Please see Major comments and this should be removed.*

**AC : Please see our answer to major comments. There was a mistake here : this is a good spatial correlation between AOD, the strong divergence regions of the BC horizontal fluxes in the FT (not PBL as written before) and the spatial distribution of the mean upward BC mass fluxes.**

---

## Referee Report (RR1)

**Review of "Cross-polar transport and scavenging of Siberian aerosols containing black carbon during the 2012 ACCESS summer campaign" by J.-C. Raut et al.**

**General comments:**

The manuscript is much improved over the previous version. I now recommend the paper for publication in ACP and suggest only minor changes before publication.

**Minor comments:**

P1L13, "very small ($< 30\%$)", remove "very".

P1L13-14, "precipitation amounts were larger (5-10 mm)" --> the accumulated precipitation amounts were larger (5-10 mm).

P25L20-23, I think that the spatial distributions of dry deposition of BC (Figure 12a) are co-located with the divergence regions of the horizontal BC flux in the PBL (Figure 7c), which may be correspond to the BC emission regions, not the upward BC flux regions and the divergence regions of the horizontal BC flux in the FT. Please check the relationship.

P27L7-8, "TEBC(z)", it is defined by 3D (x, y, z) and the authors need not to write a function of altitude z.

P28L12 and P29L5, "deposition process". It is better to replace this by "removal". In general, deposition of aerosols indicates that the aerosols are removed from the atmosphere to the surface.